# East Gobi megalake systems reveal East Asian Monsoon dynamics over the last interglacial-glacial cycle

Hongwei Li [1], Xiaoping Yang [1] ✉, Louis Anthony Scuderi [2], Fangen Hu [3], Peng Liang [1], Qida Jiang [4], Jan-Pieter Buylaert [5], Xulong Wang[6], Jinhua Du[7], Shugang Kang [6], Zhibang Ma[4], Lisheng Wang[4] & Xuefeng Wang [4]

Intense debate persists about the timing and magnitude of the wet phases in the East Asia deserts since the late Pleistocene. Here we show reconstructions of the paleohydrology of the East Gobi Desert since the last interglacial using satellite images and digital elevation models (DEM) combined with detailed section analyses. Paleolakes with a total area of 15,500 km$^2$ during Marine Isotope Stage 5 (MIS 5) were identified. This expanded lake system was likely coupled to an 800–1000 km northward expansion of the humid region in East China, associated with much warmer winters. Humid climate across the Gobi Desert during MIS 5 likely resulted in a dustier MIS 4 over East Asia and the North Pacific. A second wet period characterized by an expanded, albeit smaller, lake area is dated to the mid-Holocene. Our results suggest that the East Asian Summer Monsoon (EASM) might have been much weaker during MIS 3.

Deserts are highly sensitive to climate fluctuations and human intervention[1]. Land degradation in deserts and surrounding areas, namely desertification, directly affects some 250 million people in the developing world[2] and, on a larger scale, dust originating from deserts has global impacts on biogeochemistry and climate forcing[3]. Reconstructing the environmental history of the deserts is of great importance for understanding climate change and its impacts on desertification. As the largest inland desert on Earth, the Gobi Desert is important paleoclimatologically because of its location at the northern limit of the East Asian Summer Monsoon (EASM) and close to the center of Siberian High which controls regional circulation, i.e. the East Asian Winter Monsoon (EAWM), over most of east Asia in winter. In addition, the Gobi Desert is a major source of the dust deposits that comprise the Chinese Loess Plateau, one of the most important terrestrial climate archives[4]. In this regard, the paleoenvironmental reconstructions of the Gobi Desert are crucial for understanding regional, and even global, paleoclimate.

Although the Gobi Desert landscape was formed as early as 2.6 million years (Ma) ago as a result of the evolution of Asian topography and climate during the Cenozoic[5], detailed knowledge of its past environment is very limited. Aeolian sands and lacustrine sediments provide some evidence of past environmental conditions in the Gobi Desert[6], however, these records are primarily Holocene in age because strong deflation events have removed much of the evidence from earlier intervals. This is especially the case for the East Gobi Desert (Fig. 1). To date, the timing and magnitude of the Gobi Desert wet phases and their relationship with the EASM prior to the Holocene are still largely unresolved[7,8]. Shoreline deposits from Huangqihai Lake imply a dry MIS 3 in association with the occurrence of loess[9], while the records obtained from Wulagai Lake indicate a warm and humid

[1]Key Laboratory of Geoscience Big Data and Deep Resource of Zhejiang Province, School of Earth Sciences, Zhejiang University, Hangzhou 310058, China. [2]Department of Earth and Planetary Sciences, University of New Mexico, Albuquerque, NM 87131, USA. [3]Geographical Research Center, Yichun University, Yichun 336000, China. [4]Key Laboratory of Cenozoic Geology and Environment, Institute of Geology and Geophysics, Chinese Academy of Sciences, Beijing 100029, China. [5]Department of Physics, Technical University of Denmark, DTU-Risø Campus, Frederiksborgvej 399, 4000 Roskilde, Denmark. [6]State Key Laboratory of Loess and Quaternary Geology, Institute of Earth Environment, Chinese Academy of Sciences, Xi'an 710061, China. [7]School of Earth Science and Resources, Chang'an University, Xi'an 710054, China. ✉e-mail: xpyang@zju.edu.cn

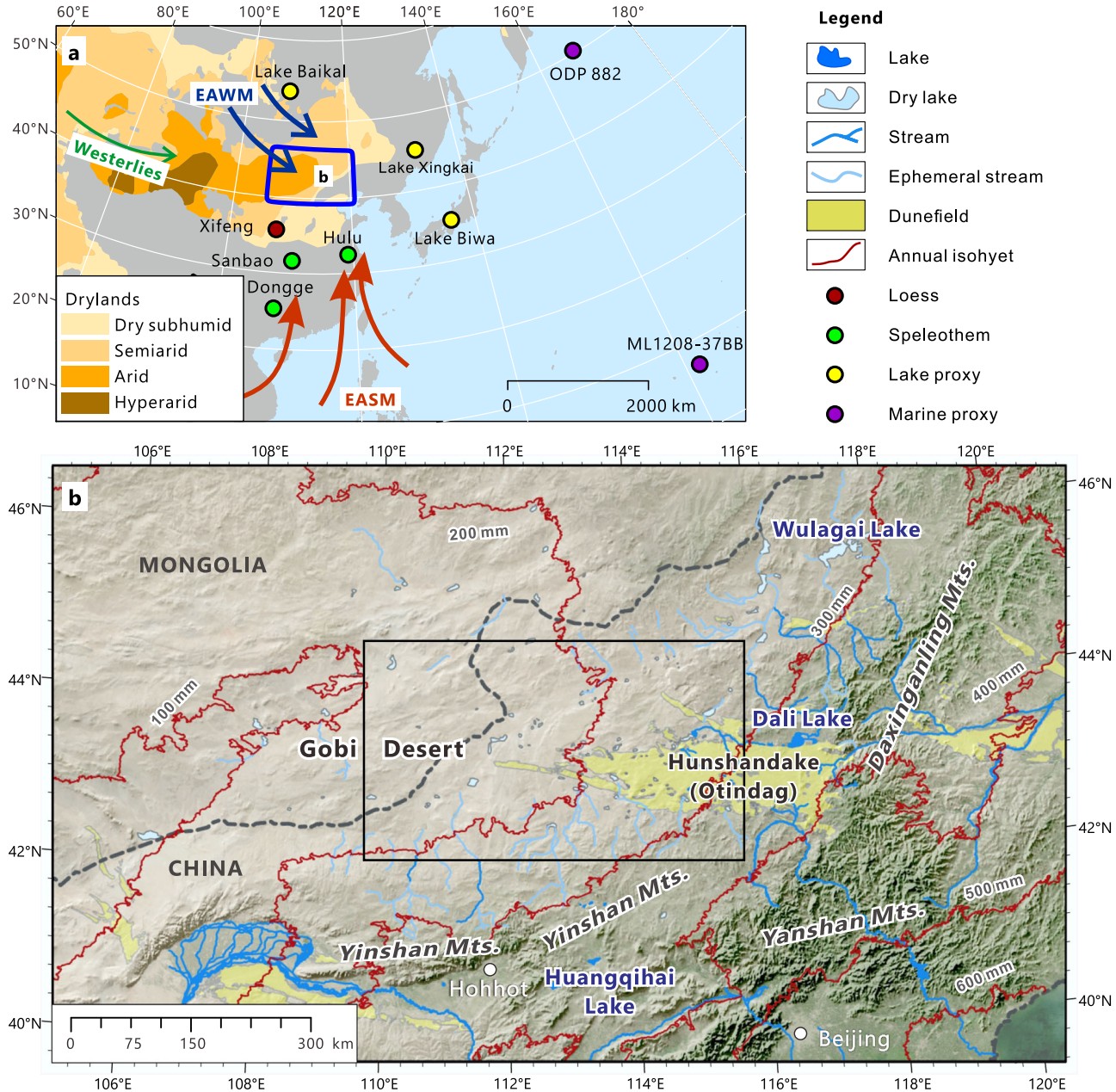

**Fig. 1 | Hydroclimatic context of the East Gobi Desert. a** Map of the East Gobi Desert showing its location within the northern mid-latitude dryland system. Prevailing winds of the East Asian Winter Monsoon (EAWM), East Asian Summer Monsoon (EASM), and the Westerlies are also indicated. Circles with different colors refer to other paleoenvironment records mentioned in the paper. **b** Topography and present drainage network of the East Gobi Desert (Drainage network data from ref. 97). Study area shown by the black rectangle. Mean annual precipitation for 1970-2000 (WorldClim Version 2[98]) is also plotted (red lines).

climate during MIS 3[10]. Additionally, the Holocene climate history of the East Gobi Desert is also in dispute. Many studies suggest that the humid period may have begun 8000 years ago[11–13]. However, other records suggest that the humid climate in this area already began in the early Holocene[9,14], and is highly correlated with southern Chinese speleothem isotope records[15]. Our experience reveals that geomorphological processes also determine the hydrological conditions in this area[16], emphasizing the importance of geomorphological studies in paleoclimate reconstruction. All these issues require further detailed investigation.

Located in the eastern part of the Asian mid-latitude desert belt, the East Gobi Desert is a relatively flat plateau bordered by mountains on the south and east (Fig. 1). Elevations decrease gradually from 1600–2000 m in the mountainous region to 800–900 m in the northwest. Due to its location near the current northern limit of the

EASM, mean annual precipitation in this area ranges from 100–400 mm with a decreasing northwestward gradient of ~60 mm/100 km (Fig. 1). Precipitation in the East Gobi Desert is characterized by a strong seasonality: summer (June–August, JJA) rainfall contributes approximately 60–70% to the annual precipitation. This combination of low precipitation and high evaporative loss produces very low runoff (<5% of the rainfall) in this area. Ephemeral streams and playas are distributed over most of the East Gobi Desert area, whereas permanent streams and lakes, albeit small and saline, are concentrated in the southeastern margin of the East Gobi Desert.

In this work, we use Digital Elevation Models (DEMs) and remote sensing images to identify the presence of a mega paleodrainage network in the East Gobi Desert. Combining this work with sedimentological evidence and dating using a combination of radiocarbon, uranium series and luminescence methods, we reconstruct East Gobi

Desert hydrological and climatic conditions over the last ~130 ka. This Gobi precipitation reconstruction differs from that interpreted from south China caves and the work also provides insights into potential causes of a "dustier MIS 4" across East Asia and the North Pacific.

## Results

### DEMs and remote sensing images identification of paleolakes and channels

As shown by DEM elevation data, there are four sub-basins, i.e. east, south, southwest, and north (hereafter designated EB, SB, SWB, and NB, respectively), in the East Gobi Desert (Fig. 2a). Analysis of a combination of remote sensing images and DEM data allowed recognition, delineation and quantification of numerous shoreline landforms in these basins.

In the SB, the most evident shoreline feature is a spit lying along the northern edge of the basin. With its highest point at ~1025 m a.s.l. (above sea level), the spit extends southeastward (~120°) ~6.5 km from a bluff (Fig. 2b). Similar shoreline features are found in the southwestern portion of the SB. A barrier system and terraces are clearly visible in the DEM data (Supplementary Fig. 1b). Elevations of the highest terrace and barrier ridge are also around 1025 m, corresponding to that of the spit. There is also a lower terrace at 1008 m. This combination of features indicates the highest paleolake level of ~1025 m, 80–90 m higher than the present floor of the basin. This paleolake level interpretation in the SB is also supported by the inflow channel (Fig. 2c). The stepped morphology shown in the longitudinal profile is probably caused by changes in the local base level, i.e. lake level in the SB. Steps in the channel bed with elevations of ~1005 m and ~1020 m (Fig. 2c) correspond with the shore terraces in the SB (Supplementary Fig. 1b), which further strengthens our interpretations.

The EB had a higher lake level than the SB. Lacustrine/fluvial deposits preserved along the south bank of a paleochannel and shore terraces suggest an EB lake level above 1040 m, ~25 m higher than the bed of the paleochannel (Fig. 2c; Supplementary Fig. 1c–e). Further DEM analysis suggests that the paleolake was not a closed system as evidenced by a 45 km long overflow channel (meandering valley) connecting the SB with the NB (Fig. 2d). The highest elevation of this channel is 1025 m, identical to that of shoreline features in the SB. A northward decrease in channel height and a delta feature at the channel interface with the NB indicates paleoflow between the SB and the NB.

Shoreline features are widespread in the NB. The longitudinal profile of the inflow meandering channel mentioned above, which has a "plateau" at 990–982 m, implies that local base level, i.e. the lake level of the NB, might have been ~990 m (Fig. 2d). Many shoreline features in the NB, including a cuspate foreland, cuspate spits, barrier systems, and shorelines all have a maximum elevation of 990 m (Fig. 2e; Supplementary Fig. 2b–d). An NB highstand is also corroborated by a fan delta that has a slope break at 990 m (Supplementary Fig. 2e, f). DEM data show that the channel connecting the NB and the SWB has a maximum elevation of ~984 m (Supplementary Fig. 2g). Therefore, the SWB merged with the NB as a single lake basin when the paleolake was at its highest level of 990 m. This observation is confirmed by a shoreline terrace-barrier complex preserved along the northern edge of the SWB (Supplementary Fig. 2g). Two terraces with altitudes of 980 m and 990 m, clearly visible in both the DEM and remote sensing imagery, correspond exactly to both the highest lake level and overflow channel that connected the SWB and NB (Supplementary Fig. 2g, h). This highest lake level in the SWB and NB is also confirmed by other shoreline features, e.g., fan deltas and shoreline scarps in the basins (Fig. 2f, g; Supplementary Fig. 2i). This combination of features suggests that the highest lake level within the SWB and NB was ~990 m. The highest lake level in the NB is also constrained by the elevation of the outlet channel situated along the western margin of the NB. This paleochannel has a maximum elevation of 990 m and exhibits a meander feature (Fig. 2h). In comparison with the channel connecting the NB and SB, which is 1200 m in width and 40 m in depth, the outflow channel of the NB is much smaller, with a channel width of ~100 m.

### Sedimentary evidence, chronology, and extent of former megalakes

In addition to the shoreline features described above, extensive sedimentary evidence also supports the contention of a large interconnected paleolake system. Sand-gravel sediments are widely distributed along the highest lake levels (Fig. 3d–f, Supplementary Figs. 3 and 4); the occurrence of symmetrical wave ripples indicates stagnant water conditions (Fig. 3d). They are consistent with each other in geomorphologic and sedimentologic expression, suggesting a lacustrine environment in this area. Shoreline deposits are consistently overlaid by 0.5–1 m of aeolian sand and sandy loess, representing the onset of desiccation and desertification. Aquatic mollusk shells, primarily Corbiculidae, are also found in shoreline deposits in the SB and the SWB, further corroborating the presence of extensive and persistent lakes in the past (Fig. 3h, Supplementary Fig. 3b).

The timing of Late Quaternary high lake levels in north China has long been contentious. In order to establish a reliable chronology, multiple dating methods were applied and evaluated in this study: radiocarbon and U-series dating of mollusk shells, U-series dating of pedogenic carbonate coatings, and luminescence dating of sandy sediment.

Four shell samples from four different sections (C, D, E, G; Supplementary Fig. 3a) were dated using radiocarbon, and the results are presented in Supplementary Table 1. Two of four shell samples gave finite radiocarbon ages of ~42 kyr BP and ~46 kyr BP, while the remaining two samples exceeded the maximum radiocarbon dating limit (Fig. 3d, Supplementary Table 1). Although in principle the radiocarbon dating range can be extended back to ~60 kyr BP[17], we suggest that radiocarbon ages greater than 40 kyr BP should be interpreted with caution. Modern carbon contamination often produces radiocarbon dating underestimates of ages older than ~30 $^{14}$C kyr BP[18]. We suspect that all the radiocarbon ages should be treated as minimum ages.

Uranium series ages of shells and their carbonate coatings are presented in Supplementary Table 2. Four U-series ages of shells fall within MIS 5, two shells yield an age consistent with the MIS 6/5 boundary, and three shells are dated to MIS 6. Three U-series shell ages are available for sections C and D (Fig. 3d). The ages of section C are consistent with those of the formation during MIS 5, whereas in section D, the shell age is ~140 kyr BP. As expected, the radiocarbon ages at these sections are much lower than the U-series ages and thus considered unrealistic. The existence of a closed system is a prerequisite for successful uranium series dating and we believe that there is evidence for open system behavior due to post-depositional recrystallization in our shell samples that could explain the older (MIS 6) ages. For example, the age of the oldest shell (sample E1 in Supplementary Table 2), uranium series dated to ~158 kyr BP, is likely overestimated. The abnormally lower content of $^{238}$U compared to shells from the same section (2586 ppb vs. 4297 ppb, Supplementary Table 2) probably suggests a loss of uranium within a non-closed system. Soluble uranium can move from the shell to a calcite coating that is formed by secondary precipitation. Indeed, we do observe a post-depositional calcite coating on sample E1 (Fig. 3h). Uranium series dating of this shell's calcite coating suggests a formation time of no later than ~56 kyr BP (sample ET in Supplementary Table 2).

We suspect that overestimation of the shell ages by the uranium series dating is commonplace, as most shell samples have a secondary carbonate coating at the inner side of the shell, albeit much thinner than for sample E1. Combined, the uranium series ages of the shells and their coatings provide an upper and lower limit to the mega lake timing, respectively. In summary, the uranium series ages on shells and their coatings suggest that the East Gobi mega-lake probably formed during MIS 5.

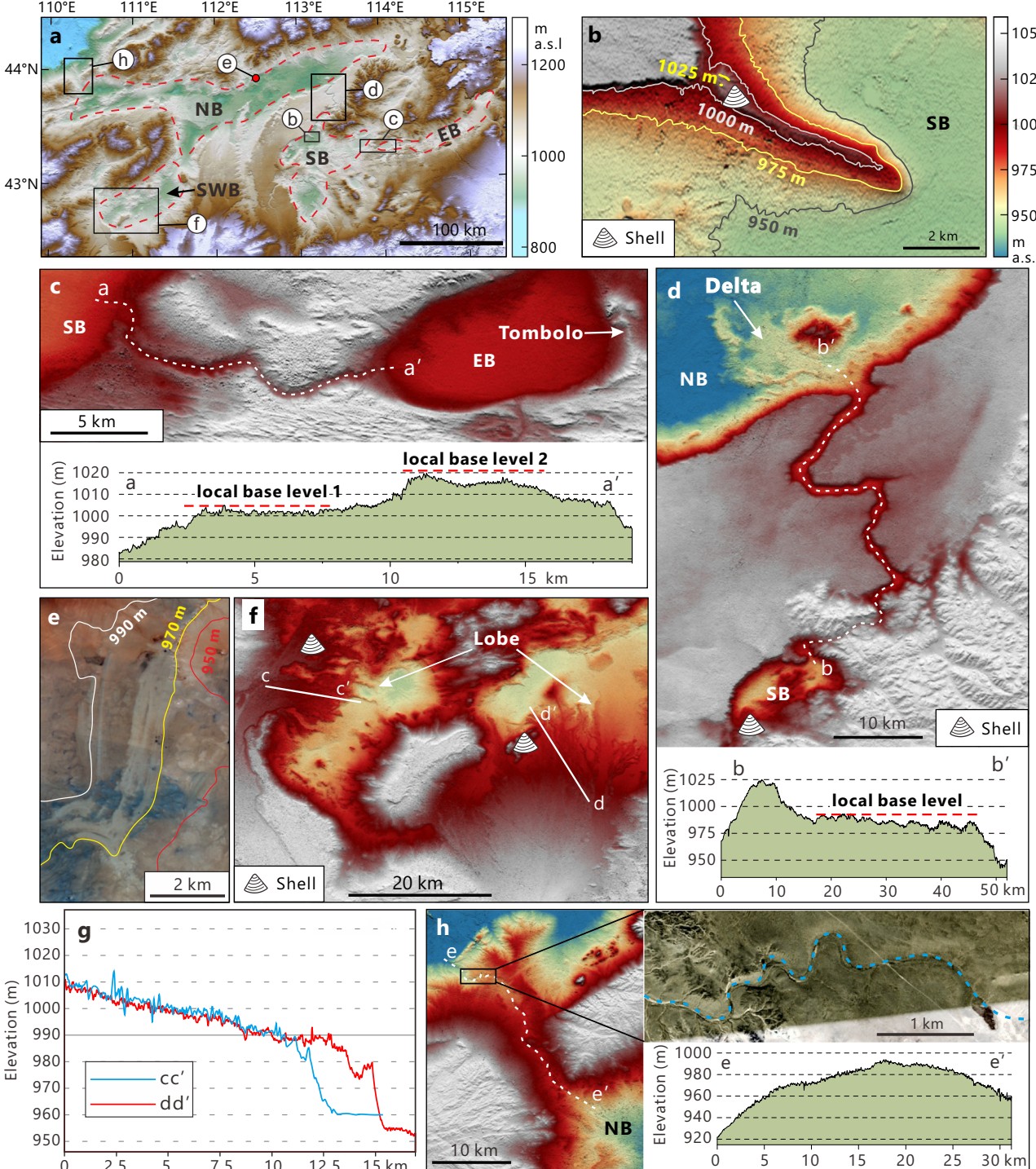

**Fig. 2 | Elevation data illustrating geomorphic evidence of a paleolake system and drainage network of the East Gobi Desert. a** Topography of the study area derived from the SRTM-3 DEM (digital elevation model, 90 m resolution). The extents of four sub-basins, i.e. eastern basin (EB), southern basin (SB), northern basin (NB), and southwestern basin (SWB) are indicated by dashed polygons. Black boxes delineate areas of higher resolution (ALOS World 3D 30 m resolution) DEM data shown in panels (**b**–**d**), (**f**) and (**h**). **b** Spit extending from the north shore of the SB. **c** DEM shows the paleochannel connecting the EB and SB. The longitudinal profile of the channel (a-a') shows a stepped-bed morphology, implying the local base levels (~1000 m and ~1020 m) change. **d** The upper panel shows a channel connecting the SB and the NB sub-basins and the delta situated at the mouth of the paleochannel. The lower panel shows the longitudinal profile (b-b') of this channel. **e** Sentinel-2A satellite image showing beach shorelines of the NB. **f** Gilbert-type fan delta feature at the southern SWB. Their cross profiles (c-c' and d-d') are shown in panel (**g**). Note that they all show a slope break at ~990 m. **h** Outflow channel of NB (left). Upper right panel- Google Earth image showing the meander feature (dashed blue line) near the mouth of the paleochannel. Lower right panel- Longitudinal profile (e-e') of the channel with a maximum elevation of ~990 m.

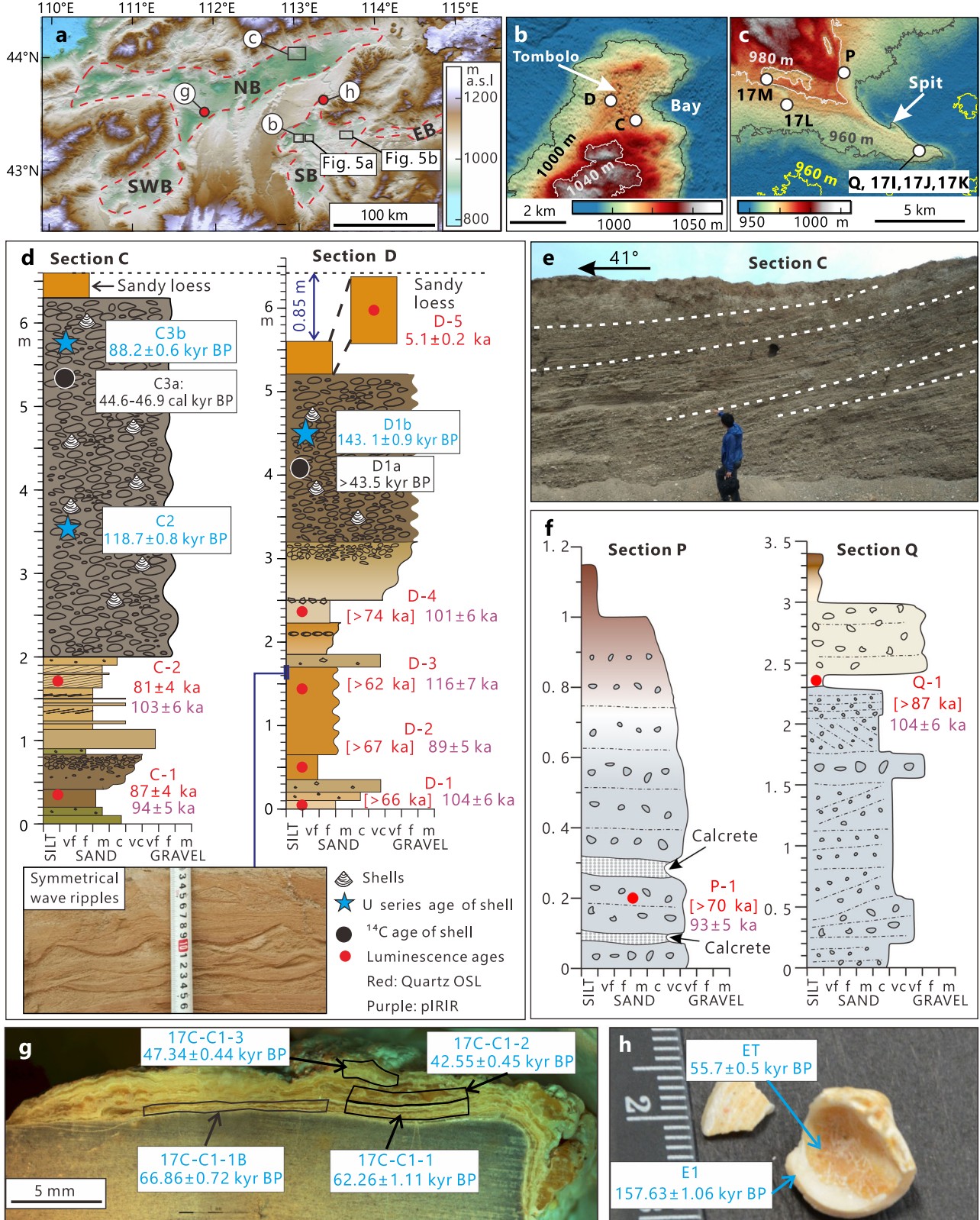

Pedogenic carbonate coatings on clasts in the NB also help to constrain the chronology of the lakes by providing minimum age estimates. Two possible models may account for the clast coating formation: dissolution of carbonate in the topsoil with downward leaching and re-precipitation in the subsoil due to water consumption or upward water movement caused by capillary rise or fluctuations of shallow groundwater[19]. In either case, the coatings are

formed after deposition, and therefore, clast coating ages should be younger than the timing of the lakes. These well-crystalized carbonate coatings have been shown to be a closed system for uranium series dating[20].

Clast carbonate coatings collected from two regions in the study area were dated using U-series. One group is from a spit along the north shore of the NB, where carbonate coatings were sampled at

**Fig. 3 | Sedimentary evidence and chronology of the East Gobi paleolake system. a** Topography of the study area derived from the SRTM-3 DEM. Black boxes delineate areas of higher resolution (ALOS World 3D 30 m) DEM data shown in panels **b**, **c** and red circles indicate sampling sites shown in panels (**g**, **h**). **b** ALOS World 3D 30 m resolution DEM showing the topography of the area around sections C and D. Sections C and D are on the east and west side of a tombolo, respectively. **c** ALOS World 3D 30 m resolution DEM showing the topography of a spit along the northern coast of the NB. The locations of the sections are indicated by white circles. **d** Stratigraphy and chronology of the beach deposits from sections C and D in the SB sub-basin. The quartz OSL (optically stimulated luminescence) ages with $D_e$ (equivalent dose) values >150 Gy are enclosed in square brackets and indicate minimum ages. **e** Photo showing the bed structure (at 2.5–6 m) of section C dipping to the NE. **f** Stratigraphy and chronology of beach deposits from sections P and Q in NB. The quartz OSL ages with $D_e$ values >150 Gy are enclosed in square brackets and indicate minimum ages. **g** Polished slab of a coastal pebble showing sampled inner rinds (outlined by the black boxes) and resulting U series ages. **h** *Corbicula fluminea* shell with yellowish carbonate coating precipitated on the inner surface of the shell visible. U series ages of the shell and its coating are also shown. Scale divisions are in mm. The photo was taken after the shell was sampled for U series dating. Note that the shell was well preserved originally with an ~2 mm thick carbonate coating inside and was broken in the laboratory for sampling. The vf very fine, f fine, m medium, c coarse, vc very coarse in panels (**d**) and (**f**) stand for the grain size of sand and gravels.

three different elevations (sections 17 K, 17 L, 17 M, Fig. 3c, Supplementary Fig. 3d–f). The other region is located in the southwestern portion of the NB (section 17 C; Fig. 3a, g, Supplementary Fig. 3c). The dense, translucent, yellow carbonate coatings yield ages ranging from the end of MIS 5 to the beginning of the Holocene, and satisfy microstratigraphic order (Fig. 3g, Supplementary Fig. 3d). The lag time between gravel deposition and the beginning of carbonate accumulation depends on factors including climate, parent material, dust input, etc. It is suggested that the lag time can be shorter than 1 ka[21]. However, more time is required to form a datable thickness of carbonate, which is a function of uranium concentration and techniques used for sampling and measurement. Recent studies suggested that the time interval from gravel deposition to the formation of datable carbonate coatings is a few thousand years[22,23]. Thus, we conclude that the oldest pedogenic coating ages (76–73 kyr BP) yield minimum age for the mega-lake system.

The luminescence dating results of lacustrine and aeolian sediments are presented in Supplementary Tables 3 and 4. Some of the quartz OSL (Optically Stimulated Luminescence) ages should be considered as minimum ages because $D_e$ (equivalent dose) values >150 Gy are measured. Despite the fact that the natural signals lie below the saturation level of the quartz dose-response curve (Supplementary Fig. 5), the accuracy of such large quartz $D_e$ values is not guaranteed, and age underestimates are often reported[24]. Indeed, when these samples are dated using post-infra-red (IR) infrared stimulated luminescence (post-IR IRSL) on K-rich feldspar extracts, significantly older ages are obtained with $D_e$ values ranging from 400–470 Gy; natural post-IR IRSL signals lie well-below the saturation level of the post-IR IRSL dose response curve (Supplementary Fig. 6, Supplementary Table 4).

Quartz OSL dating of shoreline sediments within the SB (sections C and D) indicates that their deposition likely occurred during MIS 5 (Fig. 3d); for section D only minimum ages are presented whereas for section C (quartz OSL $D_e$ values <150 Gy) the quartz OSL ages (81–87 ka) are considered finite (or possibly only slight underestimates). Minimum quartz OSL ages are also presented for samples from sections 17 N, 17I, P, Q, and 17E (Supplementary Fig. 4c, d; Supplementary Tables 3 and 4). The minimum quartz OSL age (sample 17N-1) of >90 ka for the lacustrine deposits in the paleochannel connecting the EB and the SB suggests that the 1040 m lake level of the EB occurred during MIS 5 (Supplementary Fig. 1c). K-feldspar post-IR IRSL dating was applied to the samples from sections C, D, P, Q, 17E, 17I, 19C, 19E, 19K, 19L, 19M (Fig. 3d, f; Supplementary Figs. 3a, 4, 12; Supplementary Table 4). The resulting K-feldspar ages are significantly older than the minimum quartz ages and range from 141 ± 8 ka (sample 19C-1) to 92 ± 5 ka (sample 19E-4) constraining the formation of the shoreline deposits in NB, SWB and SB to MIS 5. For the samples from sections 19K, 19L, and 19M, only minimum ages are given due to the saturation of the natural signal.

Figure 4a summarizes all the chronological information for the different elevations of the paleolake system. Taking into account the limitations of the employed dating techniques, we conclude that our dating results constrain the formation of the Eastern Gobi megalake system to MIS 5.

The megalake system is located in the northeast China fault block region, which is believed to be a tectonically stable area during the late Quaternary[25]. Earthquake records and GPS data also suggest tectonic movement and crustal deformation are negligible in the study area[26,27]. We have not found any signs of tectonic deformation from the shoreline profiles. Additionally, and more importantly, the elevation and chronology of shoreline landforms from different sites, some of which are over 200 km apart, agree well across the study area, indicating that tectonic effects are unlikely to be a factor affecting paleolake reconstruction. Estimation of the mega-lake extent based on ALOS World 3D DEM data shows that, at their maximum during MIS 5, paleolakes in the East Gobi Desert covered an area of ~15,500 km² with a maximum depth of ~90 m and a volume of 489 km³ (Fig. 4b). Humid periods probably triggered the reactivation of the river network in the East Gobi Desert. Most rivers rose from the southern mountains and flowed into the EB, SB, and SWB, some of which have a length over 200 km (Fig. 1b; Supplementary Fig. 7). Compared to other sub-basins, there are few channels found in the NB, and it was likely fed by the SB and SWB through the inter-lake channels (Supplementary Fig. 7; Fig. 4b)

We also delineated the largest lake extent in the SB during the Holocene (Figs. 4b and 5). Evidence from the SB suggests that the highest lake level during the Holocene was ~1008 m a.s.l. (Fig. 5). The terraces along the southwest shore of the SB (Supplementary Fig. 1b) and longitudinal profile of the paleochannel connecting the SB and the EB (Fig. 2c) also indicate there was a high lake level in the SB at ~1008 m, corresponding with the records in Fig. 5. The highest lake level during the Holocene is 17 m lower than the outflow channel of the SB (Fig. 2d). Therefore, the Holocene high stand in SB can be regarded as an effective indicator for peak precipitation, dated to ~6.7 ka with quartz OSL at two sections (Fig. 5c, Supplementary Table 3). The paleolakes in the mid-Holocene had a much smaller area than the lakes of MIS 5, with the highest lake levels ~17 m lower in the SB (Fig. 5b).

## Discussion

The abundance of Corbiculidae in the shoreline deposits indicates a much warmer climate during MIS 5. Originating in Eastern Asia, *Corbicula fluminea* (Müller, 1774) is a freshwater invasive species that has dispersed worldwide[28]. It is less tolerant to environmental fluctuations than other species with lower winter temperatures inducing *C. fluminea* mass mobility[29]. While it has been suggested that a coldest month mean temperature 0 °C is the lower limit of *C. fluminea* in China[8], the species is found in areas with January mean temperatures as low as −8 °C (Supplementary Fig. 8a). A database of *C. fluminea* in North America also shows a northward decrease in specimen records (Supplementary Fig. 8b) with fewer *C. fluminea* found in areas with January mean temperatures falling below −5 °C (except in artificially heated waterbodies). Further, it has been found that a warming climate in recent years has allowed a north and northwest extension of *C. fluminea* in North America, indicating

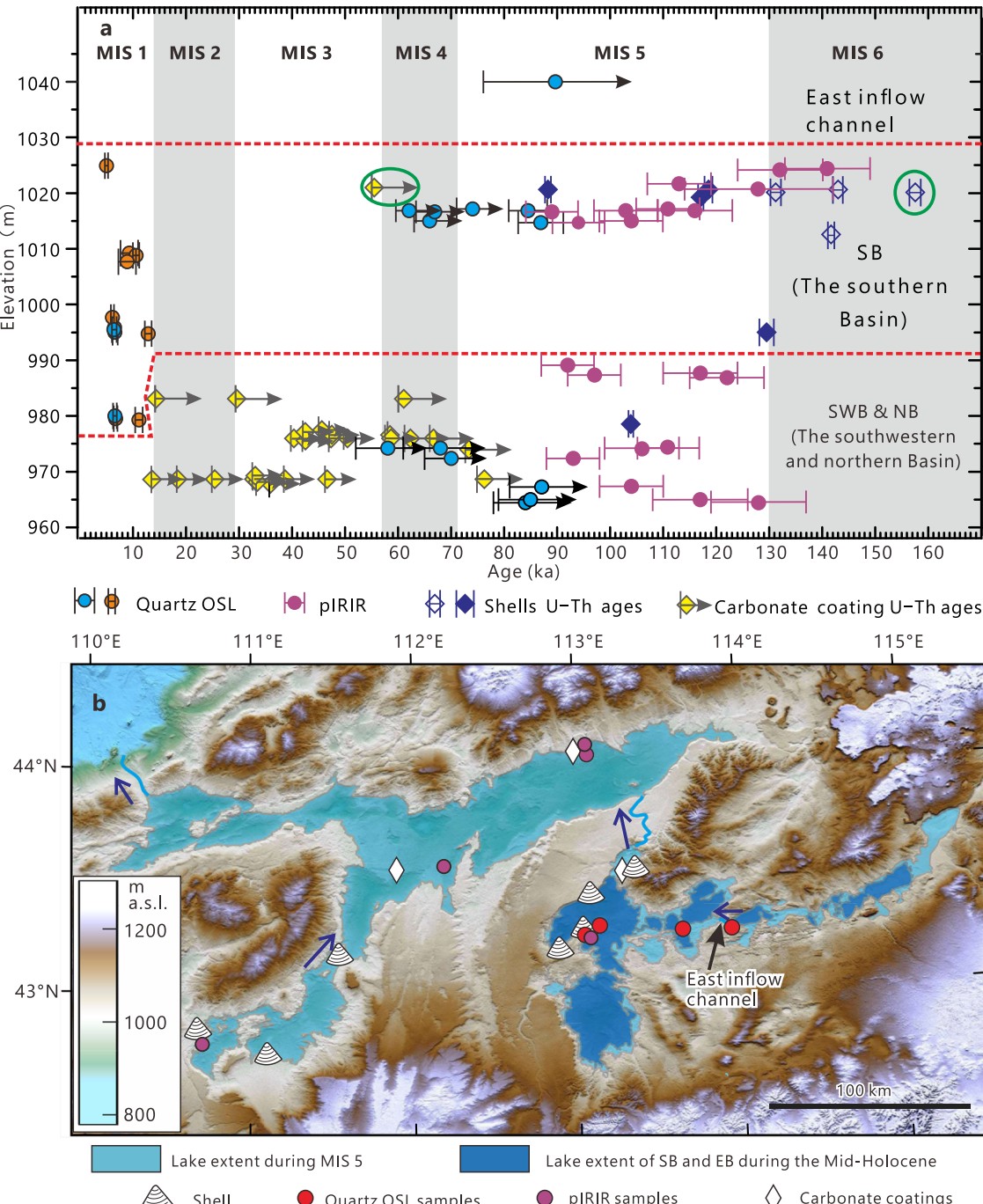

**Fig. 4 | Chronology and extent of MIS 5 and mid-Holocene paleolakes.**
**a** Paleolake age estimated by different dating methods. The blue/purple circles show the quartz (OSL)/K-feldspar (post-infra-red infrared stimulated luminescence, pIRIR) luminescence ages of shoreline deposits in the SB and NB, with the 1-sigma uncertainty represented by the vertical bars; quartz OSL ages for aeolian deposits from shorelines are represented by orange circles (Supplementary Tables 3 and 4). The arrows to the right of the blue circle indicate that these quartz OSL dates are considered to be minimum ages because D$_e$ values ≥150 Gy are measured. Dark blue and yellow diamonds represent U-Th ages of shells and pedogenic carbonate coatings, respectively (2-sigma errors indicated by left side vertical bars, Supplementary Table 2). The open diamonds indicate overestimated U-Th ages of the shells due to carbonate re-crystallization, whereas the arrows at carbonate coatings U-Th ages represent the minimum age estimation of the shoreline formation. The green circles show the age of a shell and its carbonate coating (Fig. 3h). **b** Map showing sampling sites and lake extents in the study area during MIS 5 and the mid-Holocene. The blue arrows indicate flow directions between the different basins.

that climatic variables are more important than habitat variables in controlling *C. fluminea* distribution[30].

 *C. fluminea* occurrence and abundance can be considered as a robust proxy for the temperature of the coldest month. The study area, currently strongly influenced by the Siberian anticyclone, experiences cold winters with a mean January temperature of ~ −18 °C. This suggests that the coldest month mean temperature

during MIS 5 would have been in excess of 10 °C warmer than at present, with the isotherms moving northward 400−900 km in north China (Supplementary Fig. 8a). In addition, *Corbicula largillierti* (Philippi, 1844), a dominant species by volume in the study area, also provides information on paleoclimatic conditions. At present, it occurs in the Yangtze River basin[31], 1500 km south of the study area. It has also invaded Europe, North America

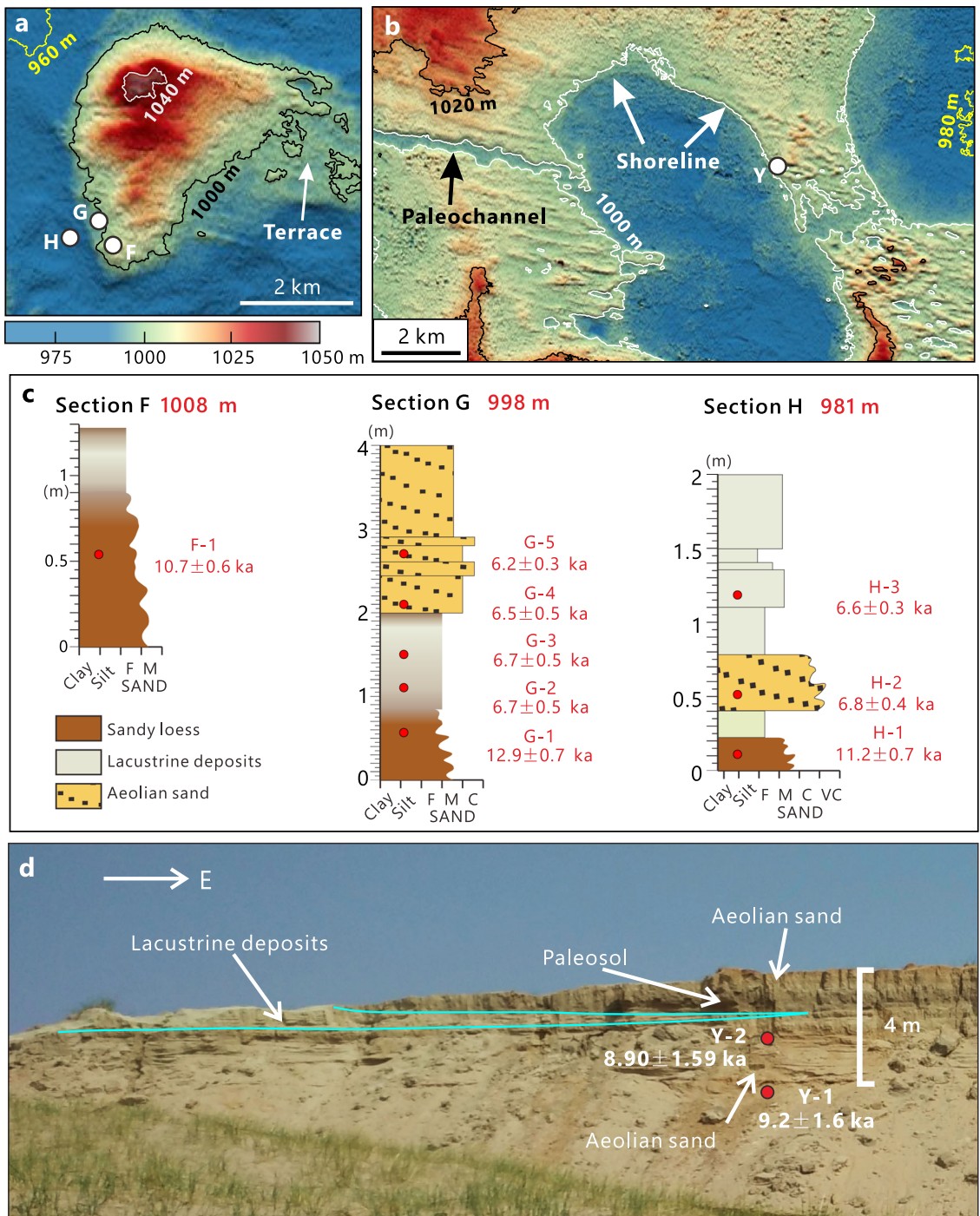

**Fig. 5 | Holocene paleolake records in the SB. a, b** ALOS AW3D DEM data showing the locations of sections and surrounding topography (Fig. 3a for the locations of the two panels). Sections F, G and H are situated on the southwest side of a former island. Lacustrine deposits have been found in all three sections and are located at different elevations. Beach ridges are visible at -1000 m in panel (**b**). **c** The stratigraphy and OSL chronology of sections F, G, and H. Note the lacustrine layer thins with increasing elevation. **d** Photo of Section Y showing its stratigraphy and OSL chronology. Note that the lacustrine layer thins to the east. The stratigraphy is in concord with its position and the surrounding topography.

and South America[32]. As with *C. fluminea*, *C. largillierti* is not reported in areas where the coldest month mean temperature falls below 0 °C.

The occurrence of *C. fluminea* also indicates low salinity water-bodies during MIS 5, as its upper lethal threshold is -8 g/kg[28]. The presence of *Parafossarulus striatulus* in the beach deposits also indi-cates a freshwater environment (Supplementary Fig. 3b). In the study area, the salinity of the lakes increases westward with decreasing

precipitation with freshwater and brackish lakes normally occurring in areas with annual precipitation greater than 300 mm. At present, the westernmost shell sites receive precipitation of only 160 mm/yr, implying that a minimum increase in annual precipitation of -100% during MIS 5 was required.

A water balance model was used to estimate precipitation during MIS 5 and mid-Holocene high stands (see methods). Due to the exis-tence of an outflow channel from the NB (the west meander, Fig. 2h),

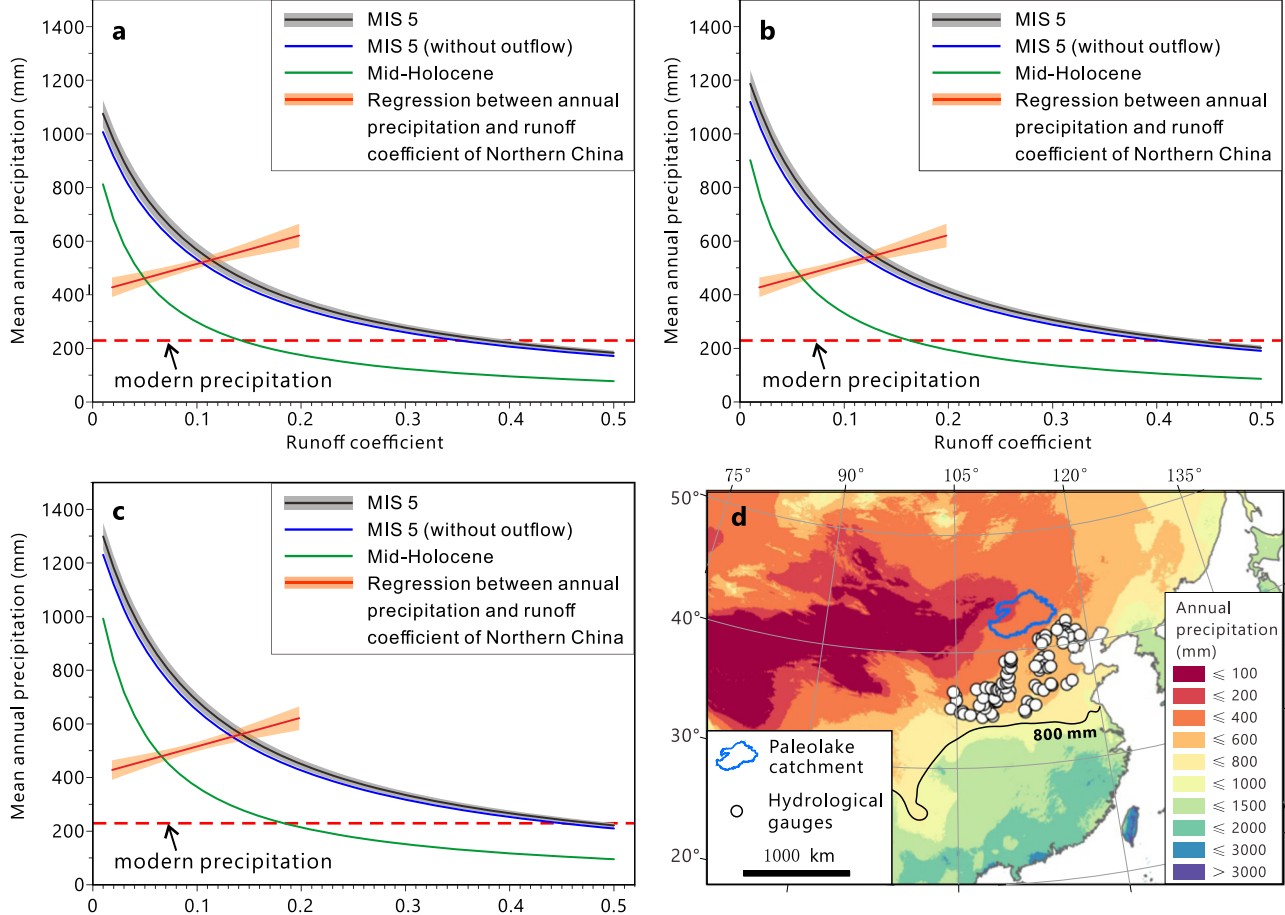

**Fig. 6 | Precipitation reconstructions. a–c** Annual precipitation of different periods estimated by a water balance model with different lake evaporation rates: **a** 900 mm/yr, **b** 1000 mm/yr, **c** 1100 mm/yr. The black and blue lines represent conditions when the NB reached its highest stand with and without outflow from the lake system through the meander (Fig. 2h), respectively. The gray shaded area shows the uncertainties caused by meander outflow volume estimation (95% confidence level) (Supplementary Fig. 9; Supplementary Table 5). The mid-Holocene precipitation is shown by the green line. The red line is the linear regression between mean annual precipitation and runoff coefficient of northern China, with the 95% confidence interval shown by the orange shade (see methods and Supplementary Fig. 10). Modern annual precipitation of the study area is ~230 mm (the red dashed line). **d** Map showing modern precipitation (Source: WorldClim Version 2.0[98]), the location of the paleolake catchment (blue framed polygon) and the hydrological gauges used for precipitation-runoff coefficient regression.

the mega-lake system during MIS 5 might not be a closed basin. However, paleochannel discharge can be estimated based on its meander geometry (Supplementary Fig. 9), and the water balance model was built to estimate precipitation during MIS 5 accounting for this outflow. As the age of the meander has not been determined, we also estimated the case when there was no outflow from the lakes. As it is shown in Fig. 6a–c, however, the influence of the west meander on the precipitation estimation is limited due to its relatively low discharge rate.

As shown in Fig. 6, our model results depend on the watershed runoff coefficient and lake evaporation rate. We selected three different lake evaporation rates in the model to estimate precipitation (see methods). The runoff coefficient always has a strong positive correlation with the mean annual precipitation[33,34]. At present, the runoff coefficient at the more humid eastern part of the study area is below 0.025[35]. We used long time series of climate and discharge data from 92 catchments, which have relatively little human interference, in North China to derive an empirical equation describing the relationship between mean annual precipitation and the runoff coefficient, which works well in the study area (Supplementary Fig. 10). Calculated study area runoff coefficients for MIS 5 and the mid-Holocene in the study area are estimated as 0.10–0.16 and 0.045–0.07, respectively (Fig. 6a–c). Using these constraints, our model results suggest that

120–150% and 80–110% more precipitation would be required to sustain these paleolakes during MIS 5 and the mid-Holocene, respectively (Fig. 6a–c). This result corresponds well with the precipitation estimated using *C. fluminea* and *Parafossarulus striatulus*. Precipitation as much as 2.2–2.5 times of modern values during MIS 5 represents a relatively stable state that produced paleolakes with high stands of 1025 m in the SB and 990 m in the NB and SWB.

At present, the 800 mm/yr isohyet is considered as the threshold of humid climate in China[36]. A 120–150% increase in mean annual precipitation during MIS 5 suggests that this humid climate limit would have moved northward 1000 km to the southern fringe of the Gobi Desert (Fig. 6d). However, the geometry of the meander valley between the NB and the SB (the east meander, Fig. 2d) suggests the discharge from the SB to the NB before meander incision would be around 36 ± 4 km³/yr (Supplementary Fig. 9), with precipitation over 1000 mm/yr (Supplementary Fig. 11). It is still not clear when the east meander formed. We interpret this marked humid period as having occurred during earlier more humid phases or during the early portion of MIS 5 when the overflow channel began to form. Considering the ~300 km³ volume of the NB (based on modern topography), an ~36 km³/yr outflow from the SB would be possible at the onset of the NB filling. Subsequently, the channel began to incise to an elevation of 990 m (i.e. the highest lake level in the NB) with annual discharge

decreasing to less than 2 km³ to sustain the high lake level, with the lower sloped connection forming a meandering channel (Fig. 2d).

The results of this study constrain major humid phases of the northern limit of the EASM since the last interglacial. Climatic conditions in northern China during the late Pleistocene have long been in debate. The timing of some lake high stands in north China and on the Tibetan Plateau, which were dated initially to MIS 3 by radiocarbon, are now arguably dated to MIS 5 by luminescence techniques. Age underestimation of older samples by radiocarbon dating is believed to be the cause of this discrepancy[7,8]. The presence of carbonate precipitates and the lack of high lake level records in our study area indicate an arid to semi-arid climate during MIS 2-4 (Fig. 7) and do not support MIS 3 as much more humid than at present in the East Gobi Desert. However, glacial and lake records from northwest China and the northwest Gobi Desert, where the climate is dominated by westerlies[37], indicate a moist MIS 3 based on multiple dating approaches[38,39]. These divergent records likely represent the different behaviors of the EASM and the westerlies during MIS 3, with the Gobi Desert serving as the boundary between these two climatic systems.

Although an extensive array of proxies has been applied in an effort to reconstruct the history and understand the mechanism of the East Asia Summer Monsoon, the timing of Holocene precipitation maximum in northern China is still controversial[12,15]. This is largely due to the lack of robust chronologies from monsoon proxies. While lake level appears to be directly related to precipitation, reliance on lake levels fails to capture the Holocene precipitation maximum in this area due to overflows caused by the humid climate[15,40]. Our results suggest that the SB was a closed-basin lake and therefore provides a reliable indicator and constraint for the Holocene optimum in the northern limit of the EASM.

Our study also provides critical information on winter temperatures during MIS 5. Proxy data indicate a much warmer climate during MIS 5, especially, for the high latitudes[41]. Palaeofloristic-based January temperature reconstructions across northern Eurasia suggest that during the last interglacial, this positive deviation gradually decreased towards the mid-latitudes, and at about 45°N the mean January temperature was close to that of the present[42]. Model results also exhibit a similar trend but the deviation is negative[43]. However, new evidence from our study suggests that significant winter warming (~10 °C greater than present) likely occurred in the East Asian mid-latitudes during MIS 5, although the duration of the warm winters remains enigmatic.

Due to its location close to the Siberian High and along the pathway of East Asian winter cold surges, the East Gobi Desert has a higher passage frequency of cold surges and a harsher winter than other areas at the same latitude[44]. Winter temperatures in the study area are therefore closely associated with the EAWM. It is possible that in the past, an EAWM with more frequent outbreaks of cold air from Siberia and Mongolia could extend its domain to the deep tropics[45]. Warmer winters during MIS 5 have also been reconstructed from records from the South China Sea[46], indicating that winter warming may expand into East Asian low latitudes due to a weakening EAWM. This pattern is quite different from the present where warming has primarily occurred in the summer and autumn in this area[47]. In recent decades, East Asia has experienced harsh winters believed to be related to a warming Arctic[48]. However, our study suggests that similar Arctic warmth might have coexisted with a warming mid-latitude East Asia during MIS 5.

The chronology of Gobi Desert paleolakes additionally provides an insight into dust emission during the late Quaternary. The Gobi Desert is a major dust source area within the northern mid-latitudes, contributing significantly to the loess deposits in China's Loess Plateau, Japan, and the Pacific, and even providing a significant component of dust in Greenland[4,49–51]. Comparison of dune sands in western China (e.g., the Taklamakan Desert and the Badain Jaran Desert) to those of the study area (i.e., the Hunshandake, Fig. 1b) shows that dune sand grain size in the study area is much coarser[52], implying a higher rate of fine sediment (dust) removal by the wind in this area. It has been suggested that the Gobi Desert contributes more dust than the Taklamakan Desert due to its relatively flat terrain and higher elevation, which is the other major dust source of East Asia[53,54]. Although dustier conditions during glacial periods have been well documented, the cause of dust flux variation at different periods is poorly constrained. Recently, growing evidence from the Pacific suggests MIS 4 was dustier than MIS 2[55,56] (Fig. 7h, i) despite MIS 2 being colder and therefore potentially dustier due to steeper meridional temperature gradients.

This interesting signal has also been captured by the records from Lake Xingkai in China, Lake Biwa in Japan, and the Chinese Loess Plateau[57–59] (Fig. 7f, g), indicating that a high dust flux during MIS 4 was a common phenomenon. We attribute the dustier MIS 4 to a higher dust supply during MIS 4 resulting from a much more humid climate across the Gobi Desert during MIS 5, when the hydrological network in the desert was active.

As clearly shown in satellite images, desert basins that filled with lakes during past wet periods (e.g., Lake Chad Basin) or which experienced alluvial processes in the Quaternary are major dust sources[60]. Field studies in the western Gobi Desert indicate that these dry lake basins currently have a dust flux roughly double that of the Gobi dune fields[61]. In addition to the mega-lake system in the East Gobi Desert, significant lake highstands also occurred in the central and west Gobi Desert during MIS 5[38,62]. Fluvial/pluvial and chemical weathering processes during a humid MIS 5 across the Gobi Desert produced more fine particles than those of the less humid MIS 3. Desiccation and shrinkage of the lakes after MIS 5 left a large area of dry lakes, alluvial fans, and riverbeds in the Gobi Desert, which likely produced a large dust flux under an intensified both EAWM regime and westerly during MIS 4.

Additionally, atmospheric circulation during glacial periods also favored the transport of dust from the Gobi Desert. During the glacial periods, the westerly jet was located to the south of the Himalaya-Tibetan Plateau throughout most of the year, and therefore, less Taklamakan Desert dust was transported out of the basin[63]. In contrast, intensified EAWM during cold periods transported more dust from the Gobi Desert to East Asia and Pacific Ocean[64,65]. Thus, the Gobi Desert was probably the major contributor to the dust over East Asia and the Pacific during MIS 2 and 4. Large-scale hydrologic activity during MIS 5 in the Gobi Desert would have induced the dustier MIS 4. This interpretation is strengthened by a nearby loess profile located downwind of the Gobi paleolake system which indicates that the loess deposition rate during MIS 4 was much higher than that of the last glacial maximum (LGM, or MIS 2)[66].

Dust flux and grain size records of MIS 2 and MIS 4 from the Chinese Loess Plateau exhibit an interesting seesaw pattern with MIS 4 being dustier than MIS 2 while the grain size of MIS 2 is coarser than that of MIS 4 (Fig. 7f). This phenomenon indicates stronger winds but a lower dust supply during MIS 2, which again supports our interpretations. The MIS 5 megalake system in the East Gobi Desert is much larger than the lakes from central and western Gobi Desert[38,62]. Therefore, it is possible that desiccation of the East Gobi megalake system produced more dust than other Gobi lakes during MIS 4. We did find unrefuted evidence for the strong deflation of the lakebed deposits in this area. As shown in Supplementary Fig. 12, luminescence dating results of the lake sediments from the lowest part of NB suggest there are no lake deposits preserved at least since MIS 6, probably indicating strong deflation in this area. Formation of the shoreline and lake-floor cliffs (Supplementary Fig. 12c, d) are also probably associated with strong deflation. Considering the high lake sedimentation rate in this area, which could reach 1.6 m/ka[35], there would have been a significant quantity of lake sediments removed by the strong winds in this area. However, a detailed provenance study is needed to quantify

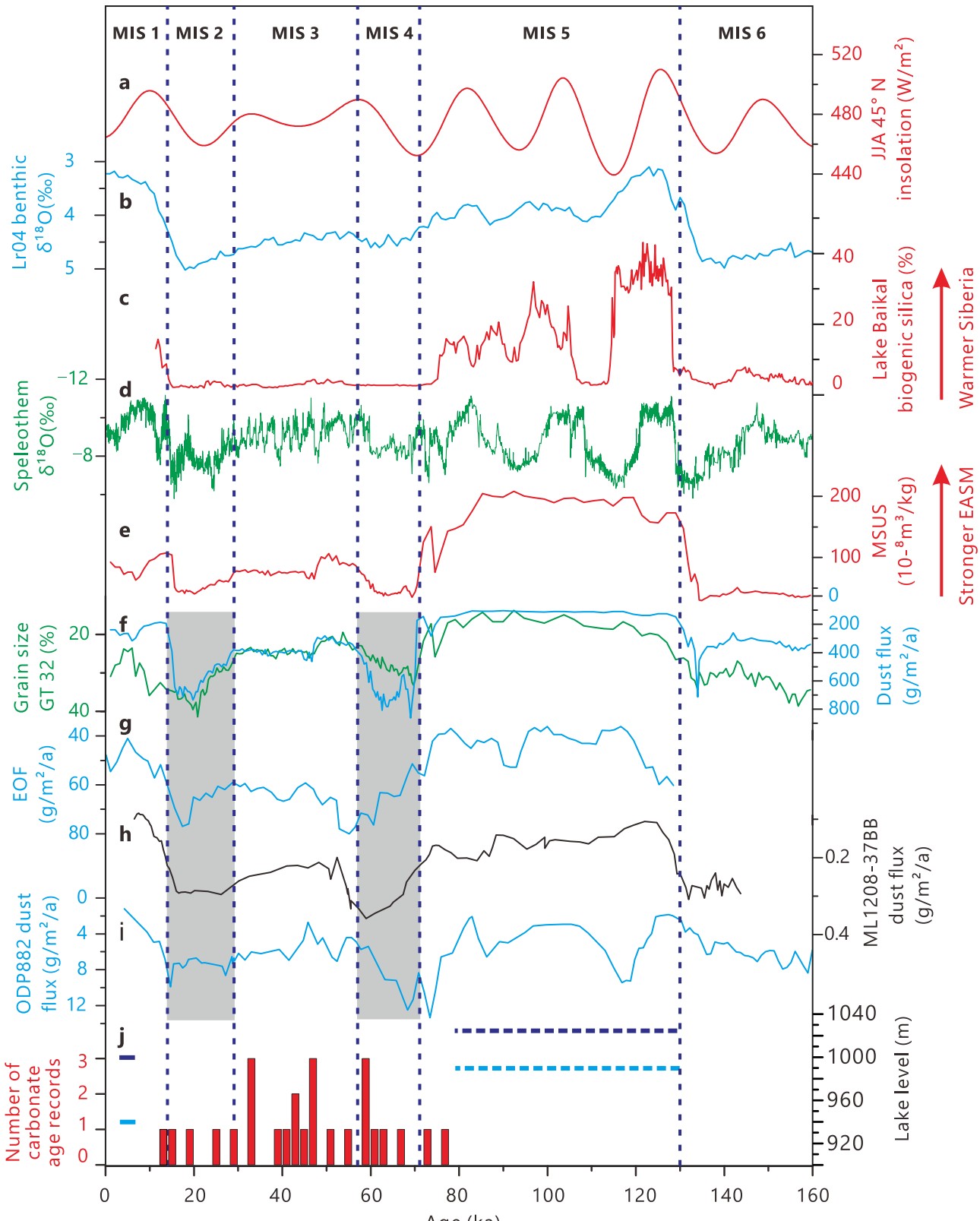

**Fig. 7 | Records from this study compared against other proxy and forcing time series. a** Summer (June to August) insolation at 45 °N[99], **b** δ[18]O record of LR04 benthic stack[100], (**c**) Lake Baikal biogenic silica record[101], **d** Composite stalagmite oxygen isotope data from south China[102]. **e, f** Magnetic susceptibility (**e**), grain size data (**f**) shown by greater than 32 μm fraction (GT 32, green) and dust flux (blue) of the loess-paleosol sequence from Xifeng, North China[57]. **g** Eolian quartz flux (EOF) from Biwa Lake sediments[59]. **h, i** Dust flux records from the Pacific[55, 56]. **j** Reconstructed lake levels from the SB (dark blue) and the NB (light blue) and frequency diagram of carbonate coating age records (red bars) as documented in this study (Fig. 1a for locations).

the contribution of the East Gobi Desert megalake system to the higher dust flux during MIS 4.

Four hypotheses have been raised to explain the dustier MIS 4 over the northern Pacific: 1) dust sources were more productive, 2) uplift and transport of dust were more efficient, 3) additional sources of dust occurred, 4) depositional processes were stronger[55]. As discussed above, the higher dust flux of MIS 4 is commonplace across East Asia and the Northern Pacific. Therefore, it requires more dust availability during MIS 4, and stronger depositional processes (hypothesis 4) alone cannot explain this phenomenon. Coarser loess deposited during MIS 2 suggests stronger winds and more efficient transport of dust than during MIS 4 (Fig. 7f). Thus, hypothesis 2 is unlikely to be the explanation for a dustier MIS 4. Considering the much wetter climate across the Gobi Desert during MIS 5, drying up of the fluvial-lake system during MIS 4 might have both increased the dust productivity and added more dust sources than MIS 2. Again, as we do not have precise provenance data, it is difficult to assess the relative contribution of hypothesis 1 and 3.

Our identification of two significant humid periods, MIS 5 and the mid-Holocene, in the East Gobi Desert corresponds well with solar insolation and marine oxygen isotope records but differs somewhat from speleothem records from south China (Fig. 7). Flow direction between channels across this 15,500 km² paleolake system can reveal the spatial characteristics of precipitation during past wet periods. The flow direction of the channel connecting SB and NB lakes suggests East Gobi Desert precipitation decreased across a northwestward gradient during MIS 5. A similar precipitation pattern is currently observed in the region and is driven by the EASM. The wet MIS 5 identified in this study was unlikely to be caused by the westerly, which typically brings moisture from the west and is characterized by an eastward decrease in precipitation. Hence, an enhanced EASM could account for the substantial increase in precipitation during MIS 5.

Speleothem $\delta^{18}O$ records from caves in southern China have been treated as robust proxies for EASM intensity and rainfall in northern China[67]. Lake levels at the northwestern limit of the EASM also appear to correlate well with cave isotope records since the last deglaciation[15]. However, based on our work, the speleothem record fails to capture marked precipitation differences during MIS 5 and the mid-Holocene at the northern continental interior limit of EASM penetration (Fig. 7). In contrast to interpretations from speleothems, the hydrologic conditions we constructed for the East Gobi Desert during these interglacials exhibit a pattern similar to those from records from the Chinese Loess Plateau (CLP) and Lake Baikal (Fig. 7). This result suggests a significant, and hitherto underappreciated, longitudinal precipitation trend in East Asia during the middle Holocene and MIS 5 that the South China speleothem record does not capture.

## Methods

### DEM and remote sensing data

GMTED2010 is global terrain elevation data developed by the U.S. Geological Survey (USGS) and National Geospatial-Intelligence Agency (NGA)[68]. The GMTED2010 used in this study has a resolution of 30 arc-seconds (about 1 km). The SRTM 90 m DEM data used in the study is v4.1 and was produced by the Consortium for Spatial Information (CGIAR-CSI) of the Consultative Group for International Agricultural Research (CGIAR)[69]. GMTED2010 and SRTM DEMs are used in smaller-scale mapping in this study for a better display. ALOS World 3D – 30 m (AW3D30) is a global digital surface model (DSM) dataset with a horizontal resolution of approx. 30-meter (1×1 arc second)[70]. It was developed by the Japanese Aerospace Exploration Agency (JAXA) using stereoscopic images acquired by the Advanced Land Observing Satellite "Daichi" (ALOS). The vertical accuracy of AW3D30 is 4.4 m (RMSE)[71]. It is suggested that AW3D30 DEM performs better in flat terrain and short vegetated areas, with the RMSE around 2 m in grassland with a gentle slope (1–5°)[72]. Our study area has a gentle slope

(mean slope of 3°) and is covered by grass. Therefore, AW3D30 DEM is suitable for geomorphological investigation in this study. Satellite images used in the study include Sentinel 2 A and Google Earth. Sentinel 2 A is a multi-spectral image data set acquired by the Multispectral Instrument (MSI) onboard the Sentinel-2 satellite[73]. The images are captured on May 26, 2016, November 25, 2016 and April 14, 2017. Visible Sentinel 2 A bands used in this study have a spatial resolution of 10 m.

### Radiocarbon dating of shells

Four Corbicula shells with clear concentric rings were selected for accelerator mass spectrometry (AMS) radiocarbon dating analysis performed at Beta Analytic Inc., Miami, USA. The samples were first washed with deionized water to remove associated organic sediments and debris. Subsequently, the samples were crushed and repeatedly subjected to HCl etches to eliminate secondary carbonate components. Calibration was completed using the INTCAL13 databases associated with 2013 INTCAL program[74,75].

### U series dating of shells and carbonate coatings

For Corbicula shells, larger pieces with clear concentric rings were selected for U series dating. The outer layers of the shells were scraped using a knife to remove debris. The cleaned inner portion was then used for further analysis. Laminated coatings on clasts are typically dense, microcrystalline carbonate, yellow to dark cream in color, ~200–4000 μm thick, with a less crystalline white-colored layer at the surface. Most of the carbonate coatings are solid and firmly adhere to the host gravels; however, a few have gaps between the laminated layers and are loosely attached to the host gravels (Supplementary Fig. 3d). Only the well crystalline carbonate coatings were selected for dating. For thick and solid coatings, we used a dental drill to collect samples from different layers. Dating was performed at the Uranium Series Dating Laboratory, Institute of Geology and Geophysics, Chinese Academy of Sciences in Beijing, China. The measurements were made on a Thermo Scientific NEPTUNE PLUS multi-collector inductively coupled plasma mass spectrometer using the procedure described in ref. 76. The procedure used to separate uranium and thorium is described in ref. 77.

### Luminescence dating

Samples for OSL dating were collected by hammering steel tubes (5 cm in diameter and 25 cm in length) into freshly cleaned section faces. Sediment in the tubes was then compacted to avoid movement and immediately sealed with opaque tape. The samples from the SB and EB were transported to the Laboratory of OSL Chronology, Institute of Earth Environment, Chinese Academy of Sciences (CAS) in Xi'an, China for dating analysis, whereas the samples from NB and Sections 19C, 19E, 19K, 19L, and 19M were measured at the Nordic Laboratory for Luminescence Dating (NLLD, Department of Physics, Technical University of Denmark, DTU Risø Campus, Denmark). Eight samples (C1, C2, D1-D4, 17I-1, and 17I-2) were dated at both laboratories. The dating results from the Xi'an lab and the NLLD are presented in Supplementary Tables 3 and 4, respectively.

In the Xi'an laboratory, the outer ~5 cm of each tube end was removed and reserved for dose rate analysis. The inner sediments were pretreated with 30% HCl and 10% $H_2O_2$ to remove carbonate and organic matter, respectively. Then, the 90–125 μm grain size fraction was sieved out and etched by HF (40%) for 45 minutes. Quartz purity was checked by IR stimulation. The Equivalent Dose ($D_e$) was determined by the Single-aliquot Regenerative-dose (SAR) protocol[78] and standard growth curves (SGC)[79] approach using a Daybreak 2200 automated OSL reader equipped with infrared (880 ± 60 nm) and blue (470 ± 5 nm) LEDs and a $^{90}S/^{90}Y$ beta source for irradiation. A preheat of 240 °C (10 s duration) and cut-heat of 220 °C (10 s) were used. Natural, regenerative, and test doses were obtained by stimulating at 125 °C

with blue LEDs for 60 s, with the OSL signal detected by an EMI 9235QA photomultiplier tube through two 3-mm U340 (290–370 nm) glass filters. Each SAR cycle ended with a blue light stimulation for 60 s (280 °C). SAR $D_e$ determination was treated with the updated method by Wang et al.[80] (Supplementary Table 3).

The environmental dose rate is an estimate of quartz grains to ionizing radiation from U-Th decay series, $^{40}K$, and cosmic sources during the burial period. Concentrations of U and Th were measured using inductively coupled plasma mass spectrometry (ICP-MS), while potassium concentration was determined by inductively coupled plasma atomic emission spectroscopy (ICP-OES). Moisture content was estimated from present field moisture conditions. The DRAC (Version 1.2) was then used to determine the environmental dose rate[81].

At the NLLD, the inner material of the tube was wet-sieved to extract the 90–180 μm and 180–250 μm grain size fractions. The fractions were treated with HCl (10%) and $H_2O_2$ (10%) to remove carbonate and organics, respectively. Then, the fractions were etched in HF (10%) to remove grain coatings and the alpha-irradiated layer from feldspar. After washing in HCl (10%), heavy liquid separation was performed using an aqueous heavy liquid (lithium heteropolytungstate, 'Fastfloat' LST) with a density of 2.58 g/ml to separate quartz and K-rich feldspar. The quartz fraction was subjected to HF (40%) treatment for 1 hour to remove the remaining feldspar and the outer alpha-irradiated rind. Both quartz and K-feldspar grains were mounted as multi-grain aliquots on stainless steel cups (~8 mm aliquots for quartz and ~2 mm for K-feldspar). Luminescence measurements were carried out on Risø TL/OSL DA-20 luminescence readers equipped with blue (470 nm; ~80 mWcm$^{-2}$) and infrared (IR, 870 nm; ~140 mWcm$^{-2}$) LEDs and β-sources calibrated for coarse-grains on stainless steel cups. Quartz luminescence was detected through a 7.5 mm U-340 Hoya filter and IR-stimulated feldspar luminescence through a blue-violet detection window made up of a combination of 2 mm BG-39 and 4 mm CN-7-59 glass filters.

The quartz extracts showed weak IR sensitivity (<10% of the corresponding blue-light stimulated luminescence) after one 60 min HF etch. Because of the limited amount of material, etching was not repeated but instead the quartz equivalent doses were measured with a "double SAR" protocol[82]. In this protocol, the aliquots are stimulated with IR for 100 s at 60 s prior to blue light stimulation to minimize the contribution from an IR sensitive signal to the blue-light stimulated luminescence from quartz. A preheat of 260 °C for 10 s and cut-heat of 220 °C were used. Natural, regenerative, and test dose signals were measured at 125 °C for 40 s. Each SAR cycle ended with a blue LEDs illumination at 280 °C (40 s). The initial 0.32 s of the blue light stimulated signal minus a background from the subsequent 0.32 s was used for $D_e$ calculation (early background subtraction)[83]. A dose recovery test was carried out on samples Q-1, 17E-1, and 17I-2 (6 aliquots per sample, given dose 100 Gy) yielding an average measured to given dose ratio of 1.20 ± 0.07. A representative dose response curve for sample 17E-1 is shown in Supplementary Fig. 5 with a natural decay curve and a calibration quartz decay curve inset. It can be seen that the quartz is dominated by a fast component and the natural signal lies well below the saturation region of the laboratory dose response curve. However, in the published literature, there is ample evidence that quartz OSL $D_e$ values > ~150 Gy are prone to underestimation[24,84]. Therefore, we only give finite quartz OSL ages for samples with $D_e$ values ≤ 150 Gy (Supplementary Tables 3 and 4).

For K-feldspar measurements, we used the same protocol as employed in Stevens et al. (2018)[85] and Yi et al. (2015)[66]: After a 320 °C preheat (60 s) treatment, natural, regenerative and test dose signals were stimulated first with IR at 200 °C for 200 s and subsequently with IR at 290 °C (200 s) (pIRIR$_{200,290}$) to access the most stable feldspar signal. The test dose was kept at ~30% of the measured dose and after each SAR cycle, a high temperature (325 °C) IR stimulation for 200 s

was used to minimize recuperation. The first 2 s of the pIRIR$_{200,290}$ decay curve minus a background derived from the last 50 s was used for calculations. A dose recovery test was carried out on seven samples (C1, D4, 17E-1, 17I-2, 19C-3, 19E-1, and 19E-4). Six aliquots per sample were bleached for 7 days in a Hönle SOL2 solar simulator. Three aliquots were used to measure the residual doses (ranging from 10 ± 2 to 28 ± 4 Gy) and three aliquots were administered a dose of 390 Gy. After subtraction of the residual dose, an average dose recovery ratio of 1.05 ± 0.03 (n = 20) was calculated indicating that our pIRIR$_{200,290}$ protocol is suitable to measure a dose given prior to heat treatment. A representative dose-response curve is shown for sample 17E-1 in Supplementary Fig. 6. It can be seen that the natural pIRIR$_{200,290}$ signal for this sample lies well below the laboratory saturation level. Typically, using pIRIR$_{200,290}$, reliable $D_e$ values up to ~900 Gy can be measured[85]. A very-hard-to-bleach but difficult-to-quantify residual dose needs to be subtracted from the pIRIR$_{200,290}$ $D_e$ values. The $D_e$ values summarized in Supplementary Table 4 have a residual of 19 ± 5 Gy (based on the residuals measured in the dose recovery test) subtracted prior to age calculation. However, because the pIRIR$_{200,290}$ $D_e$ values are so large (>400 Gy) the effect on the age is insignificant.

At the NLLD, the dose rates were measured on homogenized material using high-resolution gamma spectrometry as described in Murray et al. (2018)[86] and radionuclide concentrations (Supplementary Table 4) were converted to dry dose rates using the conversion factors listed in Guérin et al. (2011)[87]. Because the sediment was dried out upon arrival in the laboratory, the moisture content was estimated to lie half between 5% (lower limit) and saturation values measured in the laboratory (upper limit); an absolute uncertainty of ±5% was assumed on the calculated values. These estimated life-time average water content values are comparable with those estimated by the Xi'an laboratory (Supplementary Tables 3 and 4). A cosmic ray contribution was calculated and included for each sample according to Prescott and Hutton (1994)[88]. A small internal U and Th contribution to the dose rate of 0.02 ± 0.01 Gy/ka and 0.06 ± 0.03 Gy/ka was folded into the quartz and K-feldspar dose rates, respectively[89,90]. The internal $^{40}K$ beta dose rate for K-feldspar grains was calculated using a 12.0 ± 0.2% (n = 6) K concentration. This is the average concentration measured on K-rich feldspar extracts from samples P-1, Q1, 17E-1, 17E-2, 17I-1, and 17I-2 using the Risø XRF attachment (see ref. 85 for more information). A small contribution to the internal beta dose rate from Rb (assuming 400 ± 100 ppm) was also included[91]. All age calculations were performed using an in-house Excel spreadsheet ('Age calculation for coarse-grains' which is based on the age calculation spreadsheet available in Murray et al. 2021[92]).

Eight samples (C1, C2, D1-D4, 17I-1, and 17I-2) were dated using quartz OSL in both laboratories but using different luminescence instrumentation, SAR $D_e$ measurement protocols, dose rate measurement techniques, and age calculation procedures. It is reassuring to observe that a comparison of $D_e$ values, dose rates, and age results for these samples indicates that there are no systematic differences between the two laboratories (see Supplementary Tables 3 and 4).

## Water balance model to estimate paleoprecipitation

In this study, a water balance model was used to estimate precipitation during MIS 5 and the Holocene. At the steady state, the inputs of the paleolakes should equal the outputs, such that:

$$P_l A_l + c P_c A_c = E A_l + V_f + \Delta G \qquad (1)$$

where $P_l$ and $P_c$ are the precipitation amounts in the lake and catchment, respectively; $A_l$ and $A_c$ are the area of lakes and catchment (not including the lakes), respectively; $c$ is the runoff coefficient; $E$ is evaporation from the lakes; $V_f$ is outflow discharge through the paleochannel; and $\Delta G$ is groundwater discharge from the lakes and assumed to be negligible in this model. At present, the mean annual

precipitation of the MIS 5 waterbody area and catchment are 185 mm and 235 mm, respectively. We assumed the ratio of precipitation between the waterbody and catchment was constant, i.e. $P_{c(MIS\ 5)} \equiv 1.27 P_{l(MIS\ 5)}$ during MIS 5. The ratio of precipitation between the waterbody and catchment of the mid-Holocene is 1.33, which is also assumed to be constant during the mid-Holocene, i.e. $P_{c(mid\text{-}Holocene)} \equiv 1.33 P_{l(mid\text{-}Holocene)}$.

Factors affecting potential evapotranspiration (PET) include solar radiation, temperature, wind velocity, relative humidity, etc. The modern mean annual PET in the study area is 1060 mm (Supplementary Fig. 13a). Due to drier climate and higher wind speeds, desert areas in northwest China have the highest PET, which may exceed 1500 mm/yr. PET is 900–1100 mm/yr in most of South and East China (Supplementary Fig. 13a). MIS 5 has a much higher temperature than the present, which tends to increase evaporation. On the other hand, however, the wind was much weaker during MIS 5 and mid-Holocene as indicated by the grain size record of the loess deposits (Fig. 7). Also, high lake levels during MIS 5 and mid-Holocene across the study areas suggest a probably higher than the present atmospheric humidity. Both the lower wind speed and higher humidity would have offset the increasing evaporation from the warming temperature. Thus, it is difficult to estimate evaporation based only on temperature.

Considering the northward displacement of the climate zone during MIS 5 and the mid-Holocene, it is reasonable to use modern PET of East and South China, which is ~900–1100 mm/yr, as an approximation to the lake evaporation of East Gobi Desert during MIS 5 and the mid-Holocene. We believe this relatively wide range could deal with uncertainties caused by the different climate conditions. It is worth to note that some part of South China where the January temperature is 20–25 °C higher than the study area also has an annual PET of 900–1100 mm (Supplementary Fig. 13). Therefore, although the coldest month temperature of the East Gobi Desert would have been 10 °C higher than the present during MIS 5, the evaporation rate was still unlikely out of the range (i.e. 900–1100 mm/yr). In the water balance model, three different lake evaporation rates, i.e., 900 mm/yr, 1000 mm/yr and 1100 mm/yr, are used to estimate precipitation during MIS 5 and the mid-Holocene (Fig. 6a–c).

The outflow discharge ($V_f$) was estimated based on the meander geometry. The underlying regularity of meander geometry, particularly its wavelength, and its close relationship with discharge have long been recognized[93] and applied to estimate the paleodischarge of the channels on Earth and Mars[94,95]. In this study, the empirical equation between discharge and meander wavelength was developed from compiled data of modern fluvial systems in northern China (Supplementary Table 5). The meander wavelength for each site was obtained by averaging 5–8 loops from satellite imagery, whereas the river discharge data were collected from the published literature. A significant relation between meander wavelength and mean annual discharge is seen in the scatter plot (Supplementary Fig. 9). The paleodischarge of the meanders in this study (Fig. 2d, h) was estimated using the power equation shown in Supplementary Fig. 9, i.e.,

$$Q = 0.00247\lambda^{1.31912} \tag{2}$$

where Q is the annual discharge of the free meander and λ is the wavelength of the meander.

The discharge estimated by this equation is much lower than that estimated by Carlston[93], which can be ascribed to different hydrological regimes of the rivers in China and North America. Controlled by the EASM, the seasonal distribution of precipitation in China is quite uneven, with a coefficient of variation of monthly precipitation of ~50% and over 100% for the southern and northern regions of China, respectively (Supplementary Fig. 14). In contrast, monthly precipitation in the eastern region of North America, where the data used in Carlston's research were collected, is quite evenly distributed and the coefficient of variation of monthly precipitation is no greater than 30%. As a result, the ratio between mean discharge and meander channel-forming discharge, which is significantly above the mean discharge for the rivers in China, is significantly different from that calculated for North America[96].

## Data availability
The authors declare that the data generated in this study are available within the paper and its supplementary information file. GMTED2010 DEM data are available at https://topotools.cr.usgs.gov/gmted_viewer/. The SRTM 90 m DEM data are available at https://srtm.csi.cgiar.org/srtmdata/. AW3D30 DEM are available at https://www.eorc.jaxa.jp/ALOS/en/dataset/aw3d30/aw3d30_e.htm. Sentinel 2A remote sensing data are available at https://scihub.copernicus.eu/dhus/#/home. The *Corbicula fluminea* data of North America are available at https://nas.er.usgs.gov/viewer/omap.aspx?SpeciesID=92. Parts of drainage network data was applied from https://www.ngcc.cn/ngcc/html/1/index.html. *C. largillierti* data can be assessed from GBIF website: https://www.gbif.org/species/8140432. The mean annual precipitation and temperature data are available at http://www.worldclim.com/version2. The mean annual potential evapotranspiration data are available at http://thredds.northwestknowledge.net:8080/thredds/catalog/TERRACLIMATE_ALL/summaries/catalog.html?dataset=TERRACLIMATE_ALL_SCAN/summaries/TerraClimate19812010_pet.nc.

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

## Acknowledgements

We are very grateful to Yueying Liu for her help in mollusk species identification. We thank Deguo Zhang, Lydia Mackenzie, Xiao Fu, Steve Pratte, and Yiman Fang for their comments on the study. This study was funded by the National Natural Science Foundation of China (grant No. 41871007 and, 41501224, H. L.) and the Ministry of Science & Technology of China (grant No. 2017FY101001, X. Y.).

## Author contributions

H.L. and X.Y. conceived and designed the research. H.L. performed the analysis and wrote the manuscript; L.A.S. and X.Y. substantially revised it. F.H., P.L., and Q.J. contributed to the fieldwork. J.-P.B. conducted luminescence dating and revised the manuscript. X.l.W., J.D., and S.K. conducted OSL dating. Z.M., L.W., and X.f.W. contributed to the uranium series dating. All authors contributed to interpreting the data and revising the final version of the manuscript.

## Competing interests

The authors declare no competing interests.
