## [Peer Review File · Nature Communications]

East Gobi megalake systems reveal East Asian Monsoon dynamics over the last interglacial-glacial cycleReviewer #1 (Remarks to the Author):

- **What are the noteworthy results?**

The paper presents new results from the paleohydrology of the East Gobi Desert since the last interglacial. There was a larger lake system due to higher humidity especially during the MIS5. This was likely related to a 800-1000km northward expansion of more humid regions in Eastern China. These are very remarkable results.

- **Will the work be of significance to the field and related fields? How does it compare to the established literature? If the work is not original, please provide relevant references.**

The work is a significant and original contribution to the paleoecology of north-eastern China. It compares the results of the existing literature and fits to the discussion of the timing of larger lakes in the arid regions of Central Asia and the deserts of China.

- **Does the work support the conclusions and claims, or is additional evidence needed?**

Yes, the work supports the conclusion – but to be honest: the dating is still problematic. However, this is more a methodological disadvantage and not the authors fault. I think especially the problematic dating results are a disadvantage as radiocarbon data and the quartz OSL dating as both provide only minimum ages. Therefore, this largest paleolake should be a MIS 5 lake. However: Line 265: Ages 133 +/-9 are not exactly MIS 5.

Additional data are always necessary and good. In this case more dating of some of the sections will support the story much more. The timing of mega lakes in the deserts of Central Asia is also a matter of debate in the recent literature. However, in my opinion the authors provide enough evidence for the timing of the largest lake in the MIS 5. I think the evidences are clear enough.

- **Are there any flaws in the data analysis, interpretation and conclusions? Do these prohibit publication or require revision?**

Minor point: It was estimated by the authors that the time lag between the deposits of gravels and the carbonate in the system (line 240-243) is not more than a few thousand years...– This needs some more evidences, arguments or publications.

- **Is the methodology sound? Does the work meet the expected standards in your field?**

Yes. The authors using satellite images and a digital elevation model (DEM) to reconstruct the extent of a 15,500 km² large paleolake on a high standard. The dating methods are clearly described, also the water-balance model is simple but seems to be reliable.

- **Is there enough detail provided in the methods for the work to be reproduced?**
Yes.

- **Some minor comments:**

Line 93 and 98: Is the lake 60-70m or 80-90 m higher than the present floor? Please clarify.

Line 158 and subchapter: chronology and extent...: This chapter headline should be renamed as here the authors also presented the sedimentary evidences.

Frank Lehmkuhl, October 11th 2021

Reviewer #2 (Remarks to the Author):

Review of NCOMMS-21-36778 by Zhuolun Li

There is a debate regarding the timing and magnitude of the humid periods in the East Asia deserts during the late Pleistocene, which is limited to our understanding the spatial-temporal variations of the Asian monsoon. Using various dating methods, Li and his colleges confirmed the presence of a mega paleodrainage network during MIS 5 in the East Gobi Desert. Then, they reconstructed the hydrological and climatic conditions of the East Gobi Desert over the last ~130 ka and suggested that an enhanced EASM could account for increase in precipitation. From this point, this paper would be an important contribution for paleoclimate changes and dryland evolution. Moreover, this knowledge will help us to understand regional environmental changes as a response to global environmental changes. Thus, the work presented in this paper is significant scientifically.

Two important scientific findings are presented in this paper: one is that paleolakes with an area of 15,500 km² occurred at MIS 5 in the East Gobi Desert. The second is that high precipitation can be interpreted as an enhanced EASM but not by the westerlies. I believe the work is novel, and the manuscript will be of general interest and relevant to readers. I would make a few suggestions and comments on a slight revision of this manuscript as follows.

- 1. I suggest to add some details about neotectonism since MIS 5 in this region. Tectonic uplift/ subsidence can affect the altitude of paleolake level and the area of the lake.**
- 2. During MIS 5, the climate is much warmer than present. In that case, whether the expansion of the lake can be attributed to an increase in meltwater?**
- 3. For 14C dating results, I suggest to use kyr BP or yr BP to present the data. If possible, please provide PMC values in Table S1.**
- 4. In some sentences, authors use words of dry/wet to describe the climate conditions such as lines 400-421. I would personally use words of arid/humid in my own papers.**

Reviewer #3 (Remarks to the Author):

The authors provide an interesting documentation of the paleolakes system during MIS 5. However, I doubt whether the contribution represents a sufficiently striking advance to justify publication in Nature Communications. Generally, the introduction is insufficiently described, and the uncertain discussion of the methodologies and adopted datasets is lacking. I provide below line-by-line suggestions for your considerations.

Introduction: I strongly recommend you reorganize the introduction. There is no need to add too many details about the study area here; please cut it down. In contrast, the first and second paragraphs should be extended to give a more comprehensive background for leading to your contributions of this study.

Line 43. I feel disconnected here. Vegetation sensitivity is not related to the paleolakes evolution.

Line 50. I need to point out that many previous studies have discussed the paleoenvironment in the Gobi Desert. Please refer to Lu et al. (2019) Lu H, Wang X, Wang X, et al. Formation and evolution of Gobi Desert in central and eastern Asia[J]. Earth-Science Reviews, 2019, 194: 251-263.

Figure 1. There is no legend for Fig. 2b. What is the white line? What is the data source of the drainage networks (blue line).

Line 78. Change "DEM's" to DEMs; Change remote sensing data to remote sensing images. Please note that DEMs adopted in this study were also based on remote sensing methods. DEM is a kind of remote sensing data. Please modify the similar expression in the Results.

Line 78-79. There is no need to emphasize the innovation of generating the paleodrainage networks in the East Gobi Desert. It is not challenging work in the methodology.

Line 82-83. What is new light? I think the new light is your main contribution. Please add a brief explanation of your new findings here.

Line 86. What do you mean "analysis of DEM data"? It is not clear.

Line 115. Change "SRTM 90 m resolution DEM" to SRTM-3 DEM, and you can mention the resolution on the first occurrence.

Line 116. Why not only use the high-resolution AW3D30 DEM?

Figure 2 and Figure 3 are similar. I recommend merging them into one figure.

Line 158. The authors should make available full dosimetry data (age model used, total aliquots used, if any aliquots were rejected, overdispersion, grain size used for the analyses). In addition, the presentation of chronological data in the Abanico radial plot would be highly informative. Readers can then be confident in aliquot measurements, overdispersion, and robustness of the age model chosen.

Figure 7. According to Figure 7a, the calculation of precipitation follows Goldsmith et al. (2017). However, the authors just showed a picture instead of more detailed information on picking up parameters? What kind of dataset has been used? How to assess the role of temperature-induced evaporations on lake level? It is essential to make calculations more transparent. Finally, how to calculate the uncertainty of reconstructed precipitation?

Line 307. Change GW3D to AW3D

Line 376. The author listed mean annual precipitation of 376~450 mm and modern runoff coefficients of 0.10-0.15 in Luan River and Chaobai River basins to support calculated runoff coefficients (15% and 10%) for MIS 5 and the Mid-Holocene. For Dali Lake, the wetter basin, the runoff coefficient for the Mid-Holocene was estimated as 0.13, but calculated precipitation intensity up to 700~800 mm. Catchments with different dominant vegetation types probably have different runoff coefficients in the context of similar precipitation amounts. Therefore, it is unreasonable to compare runoff coefficients in different basins directly. A prominent calculation of runoff coefficients is urgent to be shown in the manuscript.

Line 459. For a high dust flux during MIS 4, the author attributed that dustier MIS 4 to a higher dust supply resulting from the desiccation of the MIS 5 Gobi megalake system. However, most literature interpreted this due to strengthened NH westerly winds. There is no doubt that the desiccation Gobi megalake system will enhance the dust accumulations. It is necessary to validate minor or major roles in dust productions in this situation. Hence, more detailed information should be implemented to discuss the high dust flux during MIS4.

Line 510. Please add the download link for all datasets. The reason for adopting each dataset should also be conveyed to readers. For example, I don't know why GMTED2010 is needed in this study.

Line 521-522. Why only the accuracy of AW3D30 is provided here, I recommend you assess the adopted DEMs by referring to the ICESat or ICESat-2. Please remove the cited conference paper.

Line 525. Please provide the collection time of the adopted images.

The last question is throughout this paper. Why do you believe that the recently collected DEMs can reveal the drainage pattern during MIS 5? Is there any possibility that the landscape evolution has reshaped river basins and changed network topology?

Response to Reviewer 1

[Comment 1]

- What are the noteworthy results?

The paper presents new results from the paleohydrology of the East Gobi Desert since the last interglacial. There was a larger lake system due to higher humidity especially during the MIS5. This was likely related to a 800-1000km northward expansion of more humid regions in Eastern China. These are very remarkable results.

- Will the work be of significance to the field and related fields? How does it compare to the established literature? If the work is not original, please provide relevant references.

The work is a significant and original contribution to the paleoecology of north-eastern China. It compares the results of the existing literature and fits to the discussion of the timing of larger lakes in the arid regions of Central Asia and the deserts of China.

Response:

We are very grateful for the reviewer's positive comments on our work.

[Comment 2]

- Does the work support the conclusions and claims, or is additional evidence needed?

Yes, the work supports the conclusion – but to be honest: the dating is still problematic. However, this is more a methodological disadvantage and not the authors fault. I think especially the problematic dating results are a disadvantage as radiocarbon data and the quartz OSL dating as both provide only minimum ages. Therefore, this largest paleolake should be a MIS 5 lake. However:
Line 265: Ages 133 +/-9 are not exactly MIS 5.

Additional data are always necessary and good. In this case more dating of some of the sections will support the story much more. The timing of mega lakes in the deserts of Central Asia is also a matter of debate in the recent literature. However, in my opinion the authors provide enough evidence for

the timing of the largest lake in the MIS 5. I think the evidences are clear enough.

Response:

Thank you for your suggestions. We acknowledge the concerns above regarding the absolute dating of the megalake. We have taken the reviewer's concern into account by dating 8 additional samples from two different sections (19E and 19C, revised Supplementary Fig. 4a, b) using K-feldspar pIRIR signals. The obtained pIRIR ages from these sections support formation during MIS 5. In addition, we have also obtained feldspar pIRIR ages for the 6 samples (2 from site C and 4 from site D, revised Fig. 3) for which originally only (slightly) underestimating quartz OSL ages were available. The pIRIR ages for sites C and D range between 89 ± 5 ka and 116 ± 7 ka, well within MIS 5, confirming our interpretation that these beach sediments were deposited during MIS 5.

With respect to the age of 133 ± 9 ka, luminescence ages are always given at 1σ and we note that even at 1σ ($133-9 = 124$ ka) the age is consistent with formation during early MIS5; at 2σ (115 ka) there is no doubt about a formation age within MIS 5. The age of this sample (17I-1) has actually been recalculated to 128 ± 9 ka due to a revised residual dose estimation based on additional bleaching experiments carried out during revision.

However, we acknowledge that also the new set of pIRIR ages is not perfect and some of the pIRIR ages probably (slightly) overestimate the time of deposition by a couple of (tens of) thousands of years, probably due to insufficient residual dose subtraction. For example, for site 19C the ages are from top to bottom 141 ± 8 ka, 132 ± 8 ka, 113 ± 6 ka and 128 ± 14 ka. In this series, the 113 ± 6 ka age is probably closest to the timing of deposition (revised Supplementary 4a). For site 19E the ages are from top to bottom: 122 ± 7 ka, 97 ± 5 ka, 117 ± 7 ka, 92 ± 5 ka (revised Supplementary 4b). So, it appears that even within a section there is greater variability in the ages than the age uncertainties can account for (over-dispersion in the data). However, it should be noted that when age uncertainties are taken into account, 19 out of the 20 pIRIR ages presented in this paper are consistent with a formation during MIS 5 (only for sample 19C-1 (141 ± 8 ka) one needs 2 standard deviations). The mean pIRIR age for all samples is 110 ± 3 ka (random error only).

We hope we further convinced the reviewer that our independent age control is strong enough to

support the conclusion that the lake formed during MIS 5.

[Comment 3]

- Are there any flaws in the data analysis, interpretation and conclusions? Do these prohibit publication or require revision?

Minor point: It was estimated by the authors that the time lag between the deposits of gravels and the carbonate in the system (line 240-243) is not more than a few thousand years...– This needs some more evidences, arguments or publications.

Response:

We have added more discussion on this issue as follows:

“The lag time between gravel deposition and the beginning of carbonate accumulation depends on factors including climate, parent material, dust input, etc. It is suggested that the lag time can be shorter than 1 ka²¹. However, more time is required to form a datable thickness of carbonate, which is a function of uranium concentration and techniques used for sampling and measurement. Recent studies suggested that the time interval from gravel deposition to formation of datable carbonate coatings is a few thousand years^{22,23}.” (*Line 245-252 of the revised manuscript*)

References:

21. Pustovoytov, K., Schmidt, K. & Parzinger, H. Radiocarbon dating of thin pedogenic carbonate laminae from Holocene archaeological sites. *Holocene* **17**, 835-843 (2007).
22. Sharp, W. D., Ludwig, K. R., Chadwick, O. A., Amundson, R. & Glaser, L. L. Dating fluvial terraces by ²³⁰Th/U on pedogenic carbonate, Wind River Basin, Wyoming. *Quat. Res.* **59**, 139-150 (2003).
23. Fletcher, K. E. K., Rockwell, T. K. & Sharp, W. D. Late Quaternary slip rate of the southern Elsinore fault, Southern California: Dating offset alluvial fans via ²³⁰Th/U on pedogenic carbonate. *J. Geophys. Res. Earth Surf.* **116**, F02006 (2011).

[Comment 4]

- Is the methodology sound? Does the work meet the expected standards in your field?

Yes. The authors using satellite images and a digital elevation model (DEM) to reconstruct the extent of a 15,500 km² large paleolake on a high standard. The dating methods are clearly described, also the water-balance model is simple but seems to be reliable.

- Is there enough detail provided in the methods for the work to be reproduced?

Yes.

[Minor comment 1]

Line 93 and 98: Is the lake 60-70m or 80-90 m higher than the present floor? Please clarify.

Response:

We apologized for this confusion. The spit is 60-70 m higher than the part of the present lake floor that is nearby the spit, while the lake level (1025 m) was 80-90 m higher than the lowest part of present floor further away from the spit. In order to avoid the confusion, we have rewritten the sentence as follows:

“With its highest point at ~1025 m a.s.l., the spit extends southeastward (~120°) ~6.5 km from a bluff (Fig. 2b).” (*Line 105-106 of the revised manuscript*)

[Minor comment 2]

Line 158 and subchapter: chronology and extent...: This chapter headline should be renamed as here the authors also presented the sedimentary evidences.

Response:

The headline has been renamed as “Sedimentary evidence, chronology and extent of former megalakes” (*Line 162 of the revised manuscript*)

Response to Reviewer 2:

[General comment]

There is a debate regarding the timing and magnitude of the humid periods in the East Asia deserts during the late Pleistocene, which is limited to our understanding the spatial-temporal variations of the Asian monsoon. Using various dating methods, Li and his colleges confirmed the presence of a mega paleodrainage network during MIS 5 in the East Gobi Desert. Then, they reconstructed the hydrological and climatic conditions of the East Gobi Desert over the last ~130 ka and suggested that an enhanced EASM could account for increase in precipitation. From this point, this paper would be an important contribution for paleoclimate changes and dryland evolution. Moreover, this knowledge will help us to understand regional environmental changes as a response to global environmental changes. Thus, the work presented in this paper is significant scientifically.

Response:

We are pleased to read that the reviewer finds our study a significant contribution to understand regional and global environmental changes.

[Comment 1]

Two important scientific findings are presented in this paper: one is that paleolakes with an area of 15,500 km² occurred at MIS 5 in the East Gobi Desert. The second is that high precipitation can be interpreted as an enhanced EASM but not by the westerlies.

I believe the work is novel, and the manuscript will be of general interest and relevant to readers. I would make a few suggestions and comments on a slight revision of this manuscript as follows.

1. I suggest to add some details about neotectonism since MIS 5 in this region. Tectonic uplift/subsidence can affect the altitude of paleolake level and the area of the lake.

Response:

We appreciate this interesting comment from the reviewer. Neotectonics would disturb paleolake shorelines and play an important role in drainage reorganization in a tectonically active region (e. g. the west Gobi Desert (Hartmann et al.,2011; Van der Wal et al.,2021; Nottebaum et al.,2022)). Our study area is located in the Northeast China fault block region, where the late Quaternary tectonic activity and seismicity are much less than in other areas in China (Deng et al.,2003). Earthquake records and GPS data also suggest tectonic movement and crustal deformation are negligible in the study area (Wang et al.,2011; Zheng et al.,2017; Wang et al.,2020). Therefore, there is only a limited number of studies on the neotectonics of this area. So, in order to assess the tectonic influence since MIS 5, we examined the shoreline features from different parts. We find that the altitudes and ages of these features correspond well with each other even over large distances. For example, the ~990 m shorelines are found both in the north and southwest basin although some of them are over 250 km apart from each other, implying no significant vertical movement in this area. Therefore, tectonic activity in this area since MIS 5 was probably (very) weak and thus is unlikely to affect our paleolake reconstruction. To address this in the main paper we have added the following paragraph:

“The megalake system is located in the northeast China fault block region, which is believed to be a tectonically stable area during the late Quaternary²⁵. Earthquake records and GPS data also suggest tectonic movement and crustal deformation are negligible in the study area²⁶⁻²⁷. We have not found any signs of tectonic deformation from the shoreline profiles. Additionally, and more importantly, the elevation and chronology of shoreline landforms from different sites, some of which are over 200 km apart, agree well across the study area, indicating that tectonic effects are unlikely to be a factor affecting paleolake reconstruction.” (*Line 282-289 of the revised manuscript*)

References

Deng Q, Zhang P, Ran Y, et al. 2003. Basic characteristics of active tectonics of China. *Science in China Series D: Earth Sciences* **46**, 356-372.

Hartmann K, Wünnemann B, Hölz S, et al. 2011. Neotectonic constraints on the Gaxun Nur inland basin in North-

Central China, derived from remote sensing, geomorphology and geophysical analyses. *Geological Society, London, Special Publications* **353**, 221-233.

Nottebaum V, Stauch G, van der Wal J L N, et al. 2022. Late Quaternary landscape evolution and paleoenvironmental implications from multiple geomorphic dryland systems, Orog Nuur Basin, Mongolia. *Earth Surf Proc Land* **47**, 275-297.

Van der Wal J L N, Nottebaum V C, Stauch G, et al. 2021. Geomorphological evidence of active faulting in low seismicity regions—examples from the Valley of Gobi Lakes, Southern Mongolia. *Frontiers in Earth Science* **8**, 589814.

Wang H, Liu M, Cao J, et al. 2011. Slip rates and seismic moment deficits on major active faults in mainland China. *Journal of Geophysical Research: Solid Earth* **116**, B02405.

Wang M, Shen Z. 2020. Present-day crustal deformation of continental China derived from GPS and its tectonic implications. *Journal of Geophysical Research: Solid Earth* **125**, e2019JB018774.

Zheng G, Wang H, Wright T J, et al. 2017. Crustal deformation in the India-Eurasia collision zone from 25 years of GPS measurements. *Journal of Geophysical Research: Solid Earth* **122**, 9290-9312.

[Comment 2]

During MIS 5, the climate is much warmer than present. In that case, whether the expansion of the lake can be attributed to an increase in meltwater?

Response:

There is no doubt that meltwater is one of the major contributors to the high lake levels during the Quaternary in the west and northwest Gobi Desert (e.g. Wünnemann et al., 2007; Lehmkuhl et al., 2018; Klinge et al., 2021; Nottebaum et al., 2022), where the mountains are much higher (~3000-4000 m) than those of the study area (1500-2000 m). However, we suggest the expansion of the lakes during MIS 5 in the East Gobi Desert cannot be attributed to an increase in meltwater. The abundance of Corbiculidae indicates a much warmer winter during MIS 5 with the winter

temperature possibly ~10°C greater than present. The highest mountain in the watershed of the paleolake is around 2000 m. Therefore, snow cover in the study area during MIS 5 would be limited if not absent. Another possible meltwater source that might have contributed to expansion of the lake is the meltwater of glaciers at the end of MIS 6. Loess records from China indicate that MIS 6 is characterized by an even dryer and colder climate than the Last Glacial Maximal (LGM) in this area (e.g. Zhang et al., 2006; Yang and Ding, 2014). However, glaciation in the drylands requires a considerable amount of precipitation (Shi, 2002; Rother et al., 2014; Lehmkuhl et al., 2016). Thus, the colder and dryer MIS 6 does not necessarily have a substantially lower equilibrium-line altitude (ELA) than the LGM. Records from the northwestern and northern margin of the Gobi Desert, which has a similar precipitation amount but a much lower temperature than the study area, suggested that ELAs during the late Pleistocene are between 2000-3000 m (Lehmkuhl et al., 2016; Khandsuren et al., 2019). It is estimated that the snowline during the LGM in the study area is around 3000 m (Shi, 2002). Considering the dryer climate during MIS 6, the ELA was unlikely to be depressed by 1000 m compared to the LGM ELA. So far, there is no Quaternary glacier recorded in the study area. In our opinion, the current understanding of Quaternary glaciations in and around the study area does not support the hypothesis that meltwater is a significant contributor to the formation of the lakes.

References:

- Khandsuren, P., Seong, Y. B., Oh, J. S., Rhee, H. H., Sandag, K., and Yu, B. Y., 2019. Late Quaternary glacial history of Khentey Mountains, Central Mongolia. *Boreas* **48**, 779-799.
- Klinge, M., Schlütz, F., Zander, A., Hülle, D., Batkhishig, O., and Lehmkuhl, F., 2021. Late Pleistocene lake level, glaciation and climate change in the Mongolian Altai deduced from sedimentological and palynological archives. *Quaternary Research* **99**, 168-189.
- Lehmkuhl, F., Grunert, J., Hülle, D., Batkhishig, O., and Stauch, G., 2018. Paleolakes in the Gobi region of southern Mongolia. *Quaternary Science Reviews* **179**, 1-23.
- Lehmkuhl, F., Klinge, M., Rother, H., and Hülle, D., 2016. Distribution and timing of Holocene and late Pleistocene glacier fluctuations in western Mongolia. *Annals of Glaciology* **57**, 169-178.

Nottebaum, V., Stauch, G., van der Wal, J. L. N., Zander, A., Schlütz, F., Shumilovskikh, L., Reicherter, K., Batkhisig, O., and Lehmkuhl, F., 2022. Late Quaternary landscape evolution and paleoenvironmental implications from multiple geomorphic dryland systems, Orog Nuur Basin, Mongolia. *Earth Surface Processes and Landforms* **47**, 275-297.

Rother, H., Lehmkuhl, F., Fink, D., and Nottebaum, V., 2014. Surface exposure dating reveals MIS-3 glacial maximum in the Khangai Mountains of Mongolia. *Quaternary Research* **82**, 297-308.

Shi, Y., 2002. Characteristics of late Quaternary monsoonal glaciation on the Tibetan Plateau and in East Asia. *Quaternary International* **97-98**, 79-91.

Wünnemann, B., Hartmann, K., Janssen, M., and Zhang Hucai, C., 2007. Responses of Chinese desert lakes to climate instability during the past 45,000 years, *Developments in Quaternary Sciences*. Elsevier, pp. 11-24.

Yang, S., and Ding, Z., 2014. A 249 kyr stack of eight loess grain size records from northern China documenting millennial-scale climate variability. *Geochemistry, Geophysics, Geosystems* **15**, 798-814.

Zhang, Z., Zhao, M., Eglinton, G., Lu, H., and Huang, C., 2006. Leaf wax lipids as paleovegetational and paleoenvironmental proxies for the Chinese Loess Plateau over the last 170kyr. *Quaternary Science Reviews* **25**, 575-594.

[Comment 3]

For ¹⁴C dating results, I suggest to use kyr BP or yr BP to present the data. If possible, please provide pMC values in Table S1.

Response:

Thank you for pointing this out. We have rewritten the ¹⁴C dating results. We have also added pMC values in Supplementary Table 1.

[Comment 4]

In some sentences, authors use words of dry/wet to describe the climate conditions such as lines 400-421. I would personally use words of arid/humid in my own papers.

Response:

We have replaced the words of dry/wet by the words of arid/humid.

Response to Reviewer 3

[General comment]

The authors provide an interesting documentation of the paleolakes system during MIS 5. However, I doubt whether the contribution represents a sufficiently striking advance to justify publication in Nature Communications. Generally, the introduction is insufficiently described, and the uncertain discussion of the methodologies and adopted datasets is lacking. I provide below line-by-line suggestions for your considerations.

Response:

We appreciate the time and effort the reviewer put into reviewing our manuscript. We have taken all the comments into account and provide a point-by-point reply to these below.

[Comment 1]

Introduction: I strongly recommend you reorganize the introduction. There is no need to add too many details about the study area here; please cut it down. In contrast, the first and second paragraphs should be extended to give a more comprehensive background for leading to your contributions of this study.

Response:

We agree with the reviewer and have revised the Introduction. Please see Page 2-3 (*Line 36-68*) of the revised manuscript.

[Comment 2]

Line 43. I feel disconnected here. Vegetation sensitivity is not related to the paleolakes evolution.

Response:

This sentence has been deleted from the manuscript.

[Comment 3]

Line 50. I need to point out that many previous studies have discussed the paleoenvironment in the Gobi Desert. Please refer to Lu et al. (2019)

Lu H, Wang X, Wang X, et al. Formation and evolution of Gobi Desert in central and eastern Asia[J]. Earth-Science Reviews, 2019, 194: 251-263.

Response:

Thank you very much for bringing the work of Lu et al. (2019) to our attention. This paper is now cited in the revised manuscript. In addition, we have also discussed and reviewed more studies in the study area. Please see *Line 50-68* of the revised manuscript.

[Comment 4]

Figure 1. There is no legend for Fig. 1b. What is the white line? What is the data source of the drainage networks (blue line).

Response:

The white lines are ephemeral streams and this has been made clear in the revised Fig. 1; we have changed the color of the ephemeral stream to light blue. The data source of the drainage networks is from the National Geomatics Center of China and our interpretation of satellite images. The data

source of the drainage network has also been added to the figure caption. (*Line 82-90* of the revised manuscript)

Revised Fig. 1. Hydroclimatic context of the East Gobi Desert. (a) Map of the East Gobi Desert showing its location within the northern mid-latitude dryland system. The prevailing winds of the East Asian Winter Monsoon (EAWM), East Asian Summer Monsoon (EASM) and the Westerlies are also indicated. Circles with different colors indicate other paleoenvironment records mentioned in the paper. (b) Topography and present drainage network of the East Gobi Desert (Drainage network data source: National Geomatics Center of China and interpretation of satellite images). Study area shown by the black rectangle. Mean annual precipitation for 1970-2000 (WorldClim Version2) is also plotted (red lines).

[Comment 5]

Line 78. Change “DEM’s” to DEMs; Change remote sensing data to remote sensing images. Please note that DEMs adopted in this study were also based on remote sensing methods. DEM is a kind of remote sensing data. Please modify the similar expression in the Results.

Response:

Thank you very much for pointing this out. All these abbreviations have been changed accordingly (*Line 91, 98, 101* of the revised manuscript).

[Comment 6]

Line 78-79. There is no need to emphasize the innovation of generating the paleodrainage networks in the East Gobi Desert. It is not challenging work in the methodology.

Response:

This sentence has been modified as follows:

“In this study, we used DEMs and remote sensing images to identify the presence of a mega paleodrainage network in the East Gobi Desert” (*Line 91-92* of the revised manuscript)

[Comment 7]

Line 82-83. What is new light? I think the new light is your main contribution. Please add a brief explanation of your new findings here.

Response:

Thank you for your suggestion. This part has been revised as follows:

“Combining this work with sedimentological evidence and dating using a combination of radiocarbon, uranium series and luminescence methods, we reconstructed the hydrological and climatic conditions of the East Gobi Desert over the last ~130 ka. This Gobi precipitation reconstruction differs from that interpreted from south China caves” (*Line 92-96* of the revised

manuscript)

[Comment 8]

Line 86. What do you mean “analysis of DEM data”? It is not clear.

Response:

This sentence has been rewritten as “As shown by DEMs data, there are four sub-basins, i.e. east, south, southwest and north (hereafter designated EB, SB, SWB and NB, respectively), in the East Gobi Desert (Fig 2a).” (*Line 99-101* of the revised manuscript)

[Comment 9]

Line 115. Change “SRTM 90 m resolution DEM” to SRTM-3 DEM, and you can mention the resolution on the first occurrence.

Response:

Thanks for your suggestions. They all have been changed accordingly. Please see *Lines 127-128, 180-181* of the revised manuscript.

[Comment 10]

Line 116. Why not only use the high-resolution AW3D30 DEM?

Response:

The reason that we used lower-resolution SRTM DEM is just for a better display. The higher

resolution DEM would produce spot-like “noise” when it is used to display a large area with the hill shading effects. It can be illustrated by the following figure:

[Comment 11]

Figure 2 and Figure 3 are similar. I recommend merging them into one figure.

Response:

We have reorganized the figures in the manuscript. We merged Fig. 2 and 3 into a new figure (*Line 125-139* of the revised manuscript), while some of the panels moved to the supplementary materials (revised Supplementary Fig. 1, 2).

[Comment 12]

Line 158. The authors should make available full dosimetry data (age model used, total aliquots used, if any aliquots were rejected, overdispersion, grain size used for the analyses). In addition, the presentation of chronological data in the Abanico radial plot would be highly informative. Readers can then be confident in aliquot measurements, overdispersion, and robustness of the age model chosen.

Response:

Total number of measured aliquots and number of rejected aliquots is now available in Supplementary Tables 3 and 4. Also the grain-size used is mentioned as foot notes below these tables or in the main text.

However, we disagree with the reviewer that over-dispersion parameters and Abanico plots would be informative. This is because both the quartz OSL and K-feldspar pIRIR measurements were made on multi-grain aliquots containing several tens to hundreds of grains. The resulting data are thus averages of the luminescence coming from different grains and therefore associated parameters (over-dispersion, skewness) do not, in our view, contain useful information. We think that dose distributions are only useful for single-grain (or in case of quartz OSL perhaps also small-aliquot data) data (e.g. Murray et al., 2021). We are not convinced that single-grain analyses of these sediments are useful given that they are from typically well-bleached environments (beach and aeolian sediments) and because of the age/dose range of the sediments under investigation. The K-feldspar pIRIR₂₉₀ De values range from ~250 to ~460 Gy which is much larger than the typical residual pIRIR₂₉₀ doses for very young or modern beach sediments of a few Gy to few tens of Gy (Murray et al., 2012; Buylaert et al., 2011, 2012; Li et al., 2015). The exact size of the hard-to-bleach residual component at time of deposition is impossible to determine and an incorrect assessment of this component for some samples (which we estimated at 19 ± 5 Gy based on laboratory bleaching experiment), may lead to overestimation for some pIRIR ages (please see also reply to the Comment 2 of reviewer 1, **Page 2-3** of the response letter). Despite these limitations our feldspar luminescence chronology confirms that the sediments under investigation were deposited during MIS 5.

References:

Buylaert, J. P., Huot, S., Murray, A. S., & Van Den Haute, P. (2011). Infrared stimulated luminescence dating of an Eemian (MIS 5e) site in Denmark using K-feldspar. *Boreas* **40**, 46-56.

Buylaert, J. P., Jain, M., Murray, A. S., Thomsen, K. J., Thiel, C., and Sohbati, R. (2012). A robust feldspar luminescence dating method for Middle and Late Pleistocene sediments. *Boreas* **41**, 435-451.

Li, G., Jin, M., Duan, Y., Madsen, D. B., Li, F., Yang, L., ... & Chen, F. (2015). Quartz and K-feldspar luminescence dating of a Marine Isotope Stage 5 megalake in the Juyanze Basin, central Gobi Desert, China. *Palaeogeography, Palaeoclimatology, Palaeoecology* **440**, 96-109.

Murray, A., Arnold, L.J., Buylaert, JP., Guérin, G., Qin, H., Singhvi, A.K., Smedley, R., and Thomsen, K.J. (2021). Optically stimulated luminescence dating using quartz. *Nat Rev Methods Primers* **1**, 72. <https://doi.org/10.1038/s43586-021-00068-5>

Murray, A. S., Thomsen, K. J., Masuda, N., Buylaert, J. P., and Jain, M. (2012). Identifying well-bleached quartz using the different bleaching rates of quartz and feldspar luminescence signals. *Radiation measurements* **47**, 688-695.

[Comment 13]

Figure 7. According to Figure 7a, the calculation of precipitation follows Goldsmith et al. (2017). However, the authors just showed a picture instead of more detailed information on picking up parameters? What kind of dataset has been used? How to assess the role of temperature-induced evaporations on lake level? It is essential to make calculations more transparent. Finally, how to calculate the uncertainty of reconstructed precipitation?

Response:

Thank you very much for your comments. We have revised this part substantially to address your concerns and hope that it is now clearer. In order to clarify the results, we have recalculated the precipitation during MIS 5 and the mid-Holocene based on the water balance model. The figures have also been reorganized, and the section of methods have been modified (*please see Line 369-407, 709-759 of the revised manuscript*).

At the steady state, the inputs of the paleolakes should equal the outputs, such that:

$$P_l A_l + c P_c A_c = E A_l + V_f + \Delta G,$$

where P_l and P_c are the precipitation amounts in the lake and catchment, respectively; A_l and A_c are the area of lakes and catchment (not including the lakes), respectively; c is the runoff coefficient; E is evaporation from the lakes; V_f is outflow discharge through the paleochannel; while ΔG is groundwater discharge from the lakes and assumed to be negligible in this model. Therefore, the uncertainties of the model results are mainly from **lake evaporation**, **outflow discharge through meander** and the **runoff coefficient** estimation, all of which will be discussed as follows:

(1) Lake evaporation

Lake evaporation is determined by many factors. In addition to the temperature, factors affecting lake evaporation also include solar radiation, wind velocity, relative humidity, etc. Many of these factors are not available for MIS 5 and therefore it is difficult to use modern evaporation models to estimate the lake evaporation during MIS 5. It has been indicated by this study that the climate zone in the study area moved northward during MIS 5 and the mid-Holocene and was characterized by an increase in both temperature and precipitation. Accordingly, evaporation rate probably also moved northward during those two interglacials across East Asia. Therefore, it would be reasonable to use modern evaporation rates in the areas that south of the East Gobi Desert as surrogates for the evaporation rates of study area during MIS 5 and mid-Holocene. There is a close relationship between lake evaporation and potential evapotranspiration (PET); it is suggested that “lake evaporation is a good surrogate for PET” (Ward et al., 2015). Thus, we used modern PET values south of the study area as proxy for the lake evaporation during MIS 5 and the mid-Holocene. As shown by the following figure, due to the dryer climate and higher wind speed, the desert area in northeast China has the highest PET, which could reach over 1500 mm/year. While the PET is 900-1100 mm/year in the most part of South and East China. Considering the northward displacement of the climate zone during MIS 5 and the mid-Holocene, it is reasonable to approximate the PET for MIS 5 and the mid-Holocene to be 900-1100 mm/year. Therefore, three different lake evaporation rates, i.e. 900 mm/year, 1000 mm/year and 1100 mm/year, are used in the model to estimate precipitation during MIS 5 and the mid-Holocene, respectively.

Revised Supplementary Fig. 12. Mean annual potential evapotranspiration (PET) (1981-2010) over the study area and surrounding regions. Data source: TerraClimate (Abatzoglou, J. et al., 2015) (*Line 149-151* of the revised supplementary materials)

(2) Outflow discharge through the paleochannels (meanders)

The outflow discharge (V_f) was estimated based on the meander geometry. The underlying regularity of meander geometry, particularly its wavelength, and its close relationship with discharge has long been recognized (Carlston, 1965) and applied to estimate the paleodischarge of the channels on earth and Mars (e.g. Burr et al., 2010; Sagri et al., 2008). In this study, the empirical equation between discharge and meander wavelength has been developed from compiled data of modern fluvial systems in north China. A significant relation between meander wavelength and mean annual discharge is seen in the following scatter plot. The 95% confidence interval of the regression was incorporated into the model uncertainties, and the error propagation

was estimated by Taylor's (1997) approach. As the age of the meanders have not been determined, we also estimated the case when there was no outflow from the lakes (See revised Fig. 6a-c, *Line 378-390* of the revised manuscript).

Revised Supplementary Fig. 9. Scatter plot of modern meander wavelength against mean annual discharge in China. Red line represents a power trendline, while the 95% confidence interval is shown by the green lines (equation shown in the plot, see Supplementary Table 5 for data). Meander channels in the study area are also shown by the dashed lines. (*Line 118-123* of the revised supplementary materials).

(3) Runoff coefficient

Runoff coefficient is affected by climate, topography (slope), vegetation coverage, catchment area, and human activity etc. (e.g. Merz and Blöschl, 2009; Cai et al., 2020; Castillo et al., 1997; Mayor et al., 2011; Ren et al., 2002). Among these factors, precipitation always shows a strong positive correlation with the runoff coefficient at areas with different climate regimes; long term runoff coefficients roughly increase with increasingly humid climates (e.g. Norbiato et al., 2009; Chang et al., 2014; Transco et al., 2016). We used long time series of climate and discharge data of 92 catchments, which have relatively few human interferences, in North China to derive an empirical equation describing the relationship between mean annual precipitation and runoff coefficient. As

is shown by the following figure, the runoff coefficients range from ~0.02 to 0.19. Although they are from areas with different geological contexts and topography, runoff coefficient exhibits a significant positive correlation with mean annual precipitation ($p < 0.0001$). We also plotted hydroclimatic conditions during the mid-Holocene from Chagan Nur (Li et al, 2020) and Dali Lake (Goldsmith et al., 2017) on this chart, both of which are located in the East Gobi Desert. As shown by the following figure, the regression line and 95% confidence level pass through the two lakes, indicating that the empirical equation works well in the East Gobi Desert. We used this empirical equation to help us determine the most possible runoff coefficients and corresponding precipitation. The confidence interval for the regression has been included in the uncertainties of precipitation reconstruction.

Revised Supplementary Fig.10. Regression between runoff coefficient (c) and mean annual precipitation (P) for 92 modern catchments of North China, with 95% confidence interval shown by the green lines. Note that the precipitation and runoff coefficients of Chagan Nur¹³ and Dali Lake¹⁴, both of which are located in the East Gobi Desert, during the Holocene high-stand periods are also plotted on this chart. The regression line and 95% confidence level pass through the two lakes, indicating the empirical equation works well in the East Gobi Desert. Data source: Yang et al. (2007) (*Line 125-132* of the revised supplementary materials)

Accordingly, the figure for the model results has also been modified as follows:

Revised Fig. 6 Precipitation reconstructions. (a-c) Annual precipitation of different periods estimated by a water balance model with different lake evaporation rates: (a) 900 mm/year, (b) 1000 mm/year, (c) 1100 mm/year. The black and blue lines represent conditions when the NB reached its highest stand with and without outflow from the lake system through the meander (Fig. 2h), respectively. The gray shaded area shows the uncertainties caused by meander outflow volume estimation (95% confidence level) (Supplementary Fig. 9; Supplementary Table 5). The mid-Holocene precipitation is shown by the green line. The red line is the linear regression between mean annual precipitation and runoff coefficient of northern China, with the 95% confidence interval shown by the orange shade (see methods and Supplementary Fig. 10). Modern annual precipitation of the study area is ~230 mm (the red dashed line). (d) Map showing modern precipitation (Source: WorldClim Version 2.0, available at <http://www.worldclim.com/version2>), the location of the paleolake catchment (blue framed polygon) and the hydrological gauges used for precipitation-runoff coefficient regression. (Line 378-390 of the revised manuscript)

References:

Abatzoglou, J. T., Dobrowski, S. Z., Park, S. A., and Hegewisch, K. C. (2018). TerraClimate, a high-resolution global dataset of monthly climate and climatic water balance from 1958–2015. *Scientific Data* **5**, 170191.

- Burr, D. M., Williams, R. M. E., Wendell, K. D., Chojnacki, M., and Emery, J. P. (2010). Inverted fluvial features in the Aeolis/Zephyria Plana region, Mars: Formation mechanism and initial paleodischarge estimates. *Journal of Geophysical Research: Planets*, **115**, E07011.
- Cai, S., Geng, H., Pan, B., Hong, Y., Chen, L. (2009). Topographic controls on the annual runoff coefficient and implications for landscape evolution across semiarid Qilian Mountains, NE Tibetan Plateau. *Journal of Mountain Science* **17**, 464–479.
- Castillo, V. M., Martinez-Mena, M., and Albaladejo, J. (1997). Runoff and soil loss response to vegetation removal in a semiarid environment. *Soil science society of America Journal*, **61**(4), 1116-1121.
- Chang, H., Johnson, G., Hinkley, T., and Jung, I. W. (2014). Spatial analysis of annual runoff ratios and their variability across the contiguous US. *Journal of Hydrology*, **511**, 387-402.
- Goldsmith, Y., Broecker, W. S., Xu, H., Polissar, P. J., Demenocal, P. B., Porat, N., ... and An, Z. (2017). Northward extent of East Asian monsoon covaries with intensity on orbital and millennial timescales. *Proceedings of the National Academy of Sciences*, **114**(8): 1817-1821.
- Li, G., Wang, Z., Zhao, W., Jin, M., Wang, X., Tao, S., ... and Madsen, D. (2020). Quantitative precipitation reconstructions from Chagan Nur revealed lag response of East Asian summer monsoon precipitation to summer insolation during the Holocene in arid northern China. *Quaternary Science Reviews*, **239**: 106365.
- Mayor, Á. G., Bautista, S., and Bellot, J. (2011). Scale-dependent variation in runoff and sediment yield in a semiarid Mediterranean catchment. *Journal of Hydrology*, **397**(1-2), 128-135.
- Merz, R., and Blöschl, G. (2009), A regional analysis of event runoff coefficients with respect to climate and catchment characteristics in Austria, *Water Resour. Res.*, **45**, W01405.
- Norbiato, D., Borga, M., Merz, R., Blöschl, G., and Carton, A. (2009). Controls on event runoff coefficients in the eastern Italian Alps. *Journal of Hydrology*, **375**(3-4), 312-325.
- Ren, L., Wang, M., Li, C., and Zhang, W. (2002). Impacts of human activity on river runoff in the northern area of China. *Journal of Hydrology*, **261**(1-4), 204-217.
- Sagri, M., Bartolini, C., Billi, P., Ferrari, G., Benvenuti, M., Carnicelli, S., and Barbano, F. (2008). Latest Pleistocene and Holocene river network evolution in the Ethiopian Lakes Region. *Geomorphology*, **94**(1-2), 79-97.

Taylor, J. R. (1997). *An introduction to error analysis, the study of uncertainties in physical measurements* (Second edition). University Science Books, Sausalito.

Trancoso, R., Larsen, J. R., McAlpine, C., McVicar, T. R., & Phinn, S. (2016). Linking the Budyko framework and the Dunne diagram. *Journal of Hydrology*, **535**, 581-597.

Ward, A. D., Trimble, S. W., Burckhard, S.R., and Lyon, J.G. (2015). *Environmental hydrology* (Third edition). CRC Press, Boca Raton, pp 121.

Yang, D., Sun, F., Liu, Z., Cong, Z., Ni, G., and Lei, Z. (2007). Analyzing spatial and temporal variability of annual water-energy balance in nonhumid regions of China using the Budyko hypothesis. *Water resources research*, **43**(4): W04426.

[Comment 14]

Line 307. Change GW3D to AW3D

Response:

Thank you for pointing out. It had been modified. (*Line 324* of the revised manuscript)

[Comment 15]

Line 376. The author listed mean annual precipitation of 376~450 mm and modern runoff coefficients of 0.10-0.15 in Luan River and Chaobai River basins to support calculated runoff coefficients (15% and 10%) for MIS 5 and the Mid-Holocene. For Dali Lake, the wetter basin, the runoff coefficient for the Mid-Holocene was estimated as 0.13, but calculated precipitation intensity up to 700~800 mm. Catchments with different dominant vegetation types probably have different runoff coefficients in the context of similar precipitation amounts. Therefore, it is unreasonable to compare runoff coefficients in different basins directly. A prominent calculation of runoff coefficients is urgent to be shown in the manuscript.

Response:

We agree with the reviewer's opinion that vegetation also affects runoff coefficient. Studies in this area suggested that vegetation shifted northward during the warm and humid interglacials (e.g. Wen et al., 2017; Tian et al., 2020). In this regard, it would be reasonable to use the modern runoff coefficients of the areas south of the study area as an analogue. We also agree with your opinion that runoff coefficients from different basins cannot be compared directly and the runoff coefficient from an individual catchment cannot be used as a surrogate to the entire study area. Therefore, we used long time series data of 92 catchments of north China (with a semi-arid to sub-humid climate that would be similar to the interglacial climates of the East Gobi Desert) to obtain a general relationship between precipitation and runoff coefficient. This would represent an average runoff coefficient value at a certain precipitation amount in north China. Our results suggest that runoff coefficients for MIS 5 and the mid-Holocene in the study area are 0.10-0.16 and 0.045-0.07, respectively. Our result for the mid-Holocene is similar with that of Chagan Nur (located at the east margin of the study area), which is estimated as ~0.041 (Li et al., 2020). We note that the estimated value is lower than that of Dali Lake which is ~0.12-0.148 for the mid-Holocene (Goldsmith et al., 2017). However, considering that the runoff coefficient increases with precipitation and Dali is located further east of the study area (much wetter than the study area), we think this difference is reasonable. In summary, our results of runoff coefficient are comparable with results of other studies around the study area, despite the use of different approaches. This, we believe, again strengthens the reliability of our results.

References:

Goldsmith, Y., Broecker, W. S., Xu, H., Polissar, P. J., Demenocal, P. B., Porat, N., ... and An, Z. (2017). Northward extent of East Asian monsoon covaries with intensity on orbital and millennial timescales. *Proceedings of the National Academy of Sciences, USA* **114**, 1817-1821.

Li, G., Wang, Z., Zhao, W., Jin, M., Wang, X., Tao, S., ... and Madsen, D. (2020). Quantitative precipitation reconstructions from Chagan Nur revealed lag response of East Asian summer monsoon precipitation to summer

insolation during the Holocene in arid northern China. *Quaternary Science Reviews* **239**, 106365.

Tian, F., Wang, Y., Zhao, Z., Li, Y., Dong, J., Liu, J., Ling, Y., Yuan, L., and Ye, M. (2020). Holocene Vegetation and Climate Changes in the Huangqihai Lake Region, Inner Mongolia. *Acta Geologica Sinica-English Edition* **94**, 1178-1186.

Wen, R., Xiao, J., Fan, J., Zhang, S., and Yamagata, H. (2017). Pollen evidence for a mid-Holocene East Asian summer monsoon maximum in northern China. *Quaternary Science Reviews* **176**, 29-35.

[Comment 16]

Line 459. For a high dust flux during MIS 4, the author attributed that dustier MIS 4 to a higher dust supply resulting from the desiccation of the MIS 5 Gobi megalake system. However, most literature interpreted this due to strengthened NH westerly winds. There is no doubt that the desiccation Gobi megalake system will enhance the dust accumulations. It is necessary to validate minor or major roles in dust productions in this situation. Hence, more detailed information should be implemented to discuss the high dust flux during MIS4.

Response:

Thank you very much for your suggestion. It is widely accepted that the westerly was strengthened and shifted southward during glacial periods due to the increased ice volume and thermal gradient (e.g. Sun, 2004; Wang et al., 2018; Abell et al., 2021). It has also been suggested that the westerly jet would locate south of Tibetan Plateau throughout most of the year during the glacial periods, and therefore less dust was transported out of Taklamakan Desert, which is the other major dust source of East Asia (Nagshima et al., 2011). During cold periods the strengthened East Asia Winter Monsoon transported more dust from the Gobi Desert to East Asia and the North Pacific (Lee et al., 2022; Kang et al., 2022). We have added more detailed information and discussion on this issue as follows:

“It is suggested that Gobi Desert contributes more dust than the Taklamakan Desert, which is the other major dust source of East Asia, due to its relatively flat terrain and higher elevation^{53,54}”

(Line 475-477 of the revised manuscript)

“We attribute the dustier MIS 4 to a higher dust supply during MIS 4 resulting from much more humid climate across the Gobi Desert during MIS 5, when the hydrological network in the desert was active” (*Line 484-486 of the revised manuscript*).

“In addition to the megalake system in the East Gobi Desert, significant lake highstands also occurred in the West Gobi Desert during MIS 5^{38,61}. Fulvial/pluvial and chemical weathering processes during humid MIS 5 in the Gobi Desert produced more fine particles than that of less humid MIS 3. Desiccation and shrinkage of the lakes after MIS 5 left large areas of dry lakes, alluvial fans and riverbeds in the Gobi Desert, which should provide a large dust flux under an intensified EAWM regime and westerly during MIS 4. Additionally, atmospheric circulation during glacial periods also favored the transport of dust from the Gobi Desert. During the glacial/stadial periods, the westerly jet located to the south of the Himalaya-Tibetan Plateau throughout most of the year, and therefore less dust from the Taklamakan Desert was transported out of the basin⁶². In contrast, intensified EAWM during cold periods transported more dust from the Gobi Desert to East Asia and Pacific Ocean^{63,64}. Thus, the Gobi Desert was probably the major contributor to the dust over East Asia and Pacific during MIS 2 and 4. Large-scale hydrologic activity during MIS 5 in the Gobi Desert would have induced the dustier MIS 4.” (*Line 491-505 of the revised manuscript*)

References

- Abell, J. T., Winckler, G., Anderson, R. F., and Herbert, T. D. (2021). Poleward and weakened westerlies during Pliocene warmth. *Nature* **589**, 70-75.
- Kang, J., Joe, Y. J., Hyun, S., Yoon, S. H., Lee, S. C., and Kim, G. Y. (2022). Origin of Asian dust reconstructed from the Eu anomaly in the Hanon paleo-maar sediment of Jeju Island, Korea. *Geo-Marine Letters* **42**, 1-10.
- Nagashima, K., Tada, R., Tani, A., Sun, Y., Isozaki, Y., Toyoda, S., and Hasegawa, H. (2011). Millennial-scale oscillations of the westerly jet path during the last glacial period. *Journal of Asian Earth Sciences* **40**, 1214-1220.
- Lee, A. M., Maruyama, A., Lu, S., Yamashita, Y., Irino, T., and Billi, A. (2022). Quantification of Asian Dust Source Variabilities in Silt and Clay Fractions since 10 Ma by Parallel Factor (PARAFAC) Endmember Modeling at IODP

Site U1425 in the Japan Sea. *Lithosphere* **2022 (Special 9)**: 6818103.

Sun, D. (2004). Monsoon and westerly circulation changes recorded in the late Cenozoic aeolian sequences of Northern China. *Global and Planetary Change* **41**, 63-80.

Wang, N., Jiang, D., and Lang, X. (2018). Northern westerlies during the Last Glacial Maximum: Results from CMIP5 simulations. *Journal of Climate* **31**, 1135-1153.

[Comment 17]

Line 510. Please add the download link for all datasets. The reason for adopting each dataset should also be conveyed to readers. For example, I don't know why GMTED2010 is needed in this study.

Response:

Thanks for pointing out. We have added the link for the datasets (*Line 554, 558, 762-775* of the revised manuscript). The reason that we used coarser resolution DEM is for a better display (please also see the response to Comment 10). The GMTED2010 is used in Fig.1 for the small-scale mapping. We have added a sentence to explain it:

“GMTED2010 and SRTM DEM are used in smaller scale mapping in this study for a better display.”
(*Line 554-555 of the revised manuscript*)

[Comment 18]

Line 521-522. Why only the accuracy of AW3D30 is provided here, I recommend you assess the adopted DEMs by referring to the ICESat or ICESat-2. Please remove the cited conference paper.

Response:

Thank you for your suggestion. The reason that we only provided the accuracy of AW3D30 is that only AW3D30 was used for geomorphological investigation in this study, whereas other two DEM datasets (GMTED2010 and SRTM) were used only for small-scale mapping. Several studies have used ICESat/ICESat-2 to evaluate AW3D30 (e.g. Li and Zhao, 2018; Li et al., 2022; Liu et al., 2020;

Takaku et al., 2020). We have cited a new peer-reviewed paper in the manuscript (Li et al., 2022) and this part has been modified as follows:

“It is suggested that AW3D30 DEM performs better in flat terrain and short vegetated area, with the RMSE around 2 m in grassland with a gentle slope (1-5°)⁶⁷. Our study area has a gentle slope (with the mean slope of 3°) and is covered by grass. Therefore, AW3D30 DEM is suitable for geomorphological investigation in this study.” (*Line 561-565 of the revised manuscript*)

Reference

Li, H., and Zhao, J. (2018). Evaluation of the newly released worldwide AW3D30 DEM over typical landforms of China using two global DEMs and ICESat/GLAS data. *IEEE Journal of Selected Topics in Applied Earth Observations and Remote Sensing* **11**, 4430-4440.

Li, H., Zhao, J., Yan, B., Yue, L., and Wang, L. (2022). Global DEMs vary from one to another: an evaluation of newly released Copernicus, NASA and AW3D30 DEM on selected terrains of China using ICESat-2 altimetry data. *International Journal of Digital Earth* **15**, 1149-1168.

Liu, Z., Zhu, J., Fu, H., Zhou, C., and Zuo, T. (2020). Evaluation of the vertical accuracy of open global dems over steep terrain regions using icesat data: A case study over hunan province, china. *Sensors* **20**, 4865.

Takaku, J., Tadono, T., Doutsu, M., Ohgushi, F., and Kai, H. (2020). Updates of 'AW3D30' ALOS global digital surface model with other open access datasets. *International Archives of the Photogrammetry, Remote Sensing & Spatial Information Sciences* **43**, 183-189.

[Comment 19]

Line 525. Please provide the collection time of the adopted images.

Response:

Thanks for pointing this out. We have added the collection time of the images:

“The images are captured on May 26, 2016, November 25, 2016 and April 14, 2017.” (*Line 568-*

[Comment 20]

The last question is throughout this paper. Why do you believe that the recently collected DEMs can reveal the drainage pattern during MIS 5? Is there any possibility that the landscape evolution has reshaped river basins and changed network topology?

Response:

Thank you for this comment. There are several potential drivers for the topographic changes and landscape evolution. The first one is neotectonic activity. Please see our reply to Comment 1 of reviewer 2 (*Page 7-8* of the response letter).

Other possible drivers for the topographic changes and landscape evolutions include fluvial and aeolian processes. All of them would modify the shoreline features, removed the shoreline sediments and change the profile of paleochannels (please see the following figure). Some of those changes bring challenges in reconstruction of paleohydrology. In order to solve these problems, we examined the DEM and satellite images carefully, combined with intensive field work and laboratory analysis. The different types of evidence, including the age control, across the study area correspond well with each other, making us very confident about the conclusions of our study.

Some evidence of the paleohydrological network reshaped by the fluvial/alluvial and aeolian processes.

Sentinel-2A satellite image shows the gullies and alluvial/fluvial channel on the paleoshoreline of the SWB (a) and sand dunes on the lake floor of the SB (b). (c) DEM shows the paleochannel connecting the EB and SB. The stepped-bed morphology of the channel (a-a') indicates the channel formed during MIS 5 (~1020 m in elevation) was partly reshaped by the mid-Holocene fluvial incision (cut to ~1000 m).

Reviewer #1 (Remarks to the Author):

I am convinced and satisfied with the response of the authors on the reviewer comments. However, as a suggestion: some of the aspects and references of the reply can be added in the paper (if not too long).

Reviewer #2 (Remarks to the Author):

The manuscript has been modified in line with previous comments (and even more). I believe that the paper is worth publishing in Nature Communications in its present form.

Reviewer #3 (Remarks to the Author):

This is a revised version of the draft that I have previously reviewed. After looking through the paper, I found the manuscript is unsuitable for the current journal's aim. Nature Communications is a multidisciplinary journal dedicated to publishing high-quality research in all areas of the biological, physical, chemical, and earth sciences. Papers published by the journal should represent important advances of significance to specialists within each field. However, this paper is not persuaded that they represent a sufficient advance in our understanding of the East Asian Monsoon dynamics over the last interglacial-glacial cycle. Based on the comments below, I don't think this paper is suitable for a high-ranking journal like Nature Communications.

From the perspective of the title, this manuscript is relative to East Asian Monsoon dynamics over suborbital and orbital timescales. However, I can't find any discussions on East Asian Monsoon dynamics in the current version. Most of the discussion is associated with lake level variations and dust provenance during MIS 4.

The high water level during MIS 5 instead of MIS 3 has been widely studied over the past two decades in northern China and southern Mongolia (e.g., Long et al., 2012, GPC; Long and Shen, 2015, SCSD; Lai et al., 2014, JOP; Li et al., 2018, GPC, Zhang et al., 2012, QG, Lehmkuhl et al., 2018, QSR). The above-mentioned literature further gave explanations of no MIS 3 high lake level because the ¹⁴C dating method covers up as far as 50 ka. For the northern expansion of EASM, Lehmkuhl et al. (2018) also put forward that EASM could reach southern and even western Mongolia based on reconstructed lake level fluctuations. Hence, the topics in this paper are not innovative. The only attractive theme is that the high dust influx during MIS 4 is a result of the drought of megalake systems after MIS 5. Actually, four possible mechanisms have been used to interpret this phenomenon 1) dust sources were more productive for other reasons (Rea, 1994), 2) the uplift and transport of dust were more efficient (McGee et al., 2010), 3) additional sources of dust contributed to tropical and NH fluxes and 4) depositional processes were stronger. Hence, more solid and direct evidence as the previous review should be implemented to support dust is mainly derived from the dry megalake system during MIS 4. Unfortunately, there is no such evidence in the manuscript to persuade me. On the other hand, authors ignored the fact that compared with areas of the whole Gobi desert, the area of megalake systems is still a very small portion of dust sources.

In reply to reviewer 3#, I really can't give positive feedback on lake evaporation. Firstly, the authors proposed that modern evaporation rates in the south of the East Gobi Desert areas can be surrogates for the evaporation rates of the study area during MIS 5 and mid-Holocene. However, the coldest month means temperature during MIS 5 would have been over 10 °C warmer than at present based on temperature reconstructions. Such large-amplitude changes in temperature certainly have a substantial influence on lake evaporation during MIS 5 relative to MIS 1 and current conditions. Secondly, the authors inferred evaporation rate probably also moved northward during those two interglacials across East Asia. In Fig. 12, distributions of PET over China didn't favor a northward displacement of evaporation rate when precipitation intensity moved northward. This is

because there is no regular pattern of evaporation rate fluctuations in China.

Line 28, explain the corresponding period, whole MIS 5, or specific period.

Line 34-35, the Authors emphasize that their study is different from a speleothem from caves in southern China. Why is specially referred to as southern China instead of the whole of China? There also have been reported speleothem caves records in northern and southwestern China during MIS 5.

Line42-44, how much area of the Gobi Desert?

Line 64-65, the Authors mean geomorphological process rather than monsoon precipitation influenced hydrological conditions.

Line 82-83, in Fig. 1a, three red arrows array seemingly indicate the moisture is from Bengal bay. If so, it is ISM, not EASM.

Line 466-468, As authors firstly cited Kug et al. (2015), a warming Arctic is responding to harsh winters in the East Asia area. However, the authors then proposed that Arctic warmth might have coexisted with warming water based on fluminea occurrence and abundance at mid-latitude East Asia during MIS 5. This is illogical.

Line 411-412, please add a reference

The data for regression between runoff coefficient and mean annual precipitation for 92 modern catchments are from north of China where is outside the study area (Fig. 6d). It is reasonable for me to infer that regression results can't reflect the true conditions for the study area.

**Responses (in blue, line numbers in the word file of revised version are included)
to Comments (in black)**

Reviewer #1 (Remarks to the Author):

[General Comment]

I am convinced and satisfied with the response of the authors on the reviewer comments. However, as a suggestion: some of the aspects and references of the reply can be added in the paper (if not too long).

Response:

Thanks for your comments. We are grateful for your positive feedback. We agree with you that some of the content of the reply is helpful for understanding this paper. We have added more text, some of it is from the reply (*Line 760-778 of the revised manuscript*). However, as you have noted, we cannot add much as we prefer to keep the manuscript focused on its research aims.

Reviewer #2 (Remarks to the Author):

[General Comment]

The manuscript has been modified in line with previous comments (and even more). I believe that the paper is worth publishing in Nature Communications in its present form.

Response:

Thanks for your comments. We appreciate your positive feedback.

Reviewer #3 (Remarks to the Author):

[General Comment]

This is a revised version of the draft that I have previously reviewed. After looking through the paper, I found the manuscript is unsuitable for the current journal's aim. Nature Communications is a multidisciplinary journal dedicated to publishing high-quality research in all areas of the biological, physical, chemical, and earth sciences. Papers published by the journal should represent important advances of significance to specialists within each field. However, this paper is not persuaded that they represent a sufficient advance in our understanding of the East Asian Monsoon dynamics over the last interglacial-glacial cycle. Based on the comments below, I don't think this paper is suitable for a high-ranking journal like Nature Communications.

Response:

We appreciate your time, patience and feedback on our paper. We may point out that the middle part of your general comment should have quotation marks because it was copied from NC's homepage. On the basis of our evaluations, we respectfully disagree with your comment that the paper does not represent a sufficient advance in our understanding of the East Asian Monsoon dynamics over the last interglacial-interglacial cycle. We believe that this study makes several valuable contributions to this field. Specifically:

1) It provides the longest record of paleohydrology in the East Gobi Desert based on a detailed geomorphic investigation and a robust chronological control. Our study, unlike others, shows a large reconstructed lake system in the Gobi Desert during MIS 5.

2) Wet phases since the last interglacial are defined in the margin of East Asia summer Monsoon. For the first time, this paper quantitatively reconstructs the precipitation of the Gobi Desert during the last two interglacials. Seasonable temperature, i.e. coldest month temperature, of MIS 5 has also been estimated, which have great implications on the present warming climate.

3) The wet phases recognized in this study differ from the south China cave records, which has long been used as the proxy for the east Asian summer monsoon. This raises questions about the applicability of $\delta^{18}\text{O}$ from south China caves to distant systems like the Gobi, and, as we note in the paper, this may suggest previously unrecognized factors in the climate regime of this area over the past two interglacials.

4) This paper provides insights for understanding the “dustier MIS 4” across East Asia and North Pacific.

In this letter, we have carefully provided a point-by-point reply to your comments. The paper has also been revised. Hopefully this response addresses your concerns.

[Comment 1]

From the perspective of the title, this manuscript is relative to East Asian Monsoon dynamics over suborbital and orbital timescales. However, I can't find any discussions on East Asian Monsoon dynamics in the current version. Most of the discussion is associated with lake level variations and dust provenance during MIS 4.

Response:

Thanks for your comments. As our title suggests, this paper is not only relevant to understanding the East Asian Summer Monsoon dynamics, but also with the megalake systems reconstruction, They are interconnected and of equal importance in this paper. In our opinion this requires discussions of EASM dynamics, megalake system reconstructions and dust provenance.

We agree that while paleohydrology is discussed at length in this paper we believe it is necessary to do so to capture the complexity of this system. This is because, as shown in the paper, what we reconstructed is a complex drainage network comprising four lake basins and channels over a large area (with a catchment ca. 150,000 km²), not just lake level variations of

a single lake. In order to constrain the extent of the megalake system, we explored and examined extensive evidence of high lake level across the study area; this requires more than a cursory discussion of each of the elements of the system. Additionally, discussing the challenges of dating the late Pleistocene (especially the last interglacial), which was flagged by Reivewer 1 during the first round review, also requires significant discussion.

The discussion of the dust, based on suggestions made by Reviewer 3, added 215 words during the first revision according to Reviewer 3's comments. And it has increased from 641 to 985 words during this revision in order to address your concerns on this issue.

We respectfully disagree with the reviewer that there is no discussion on East Asian Monsoon dynamics. Apart from the last two paragraphs of the main text (*Line 516-549 of the last round revised manuscript; Line 543-576 of this round revised manuscript*), another 1700 words in the discussion section were used to illustrate precipitation and winter temperature reconstructions, both of which are indispensable part of understanding the East Asian Monsoon (*Line 336-469 of the last round revised manuscript; Line 336-470 of this round revised manuscript*). In total, there are over 2000 words in the discussion of this topic.

[Comment 2]

The high water level during MIS 5 instead of MIS 3 has been widely studied over the past two decades in northern China and southern Mongolia (e.g., Long et al., 2012, GPC; Long and Shen, 2015, SCSD; Lai et al., 2014, JOP; Li et al., 2018, GPC, Zhang et al., 2012, QG, Lehmkuhl et al., 2018, QSR). The above-mentioned literature further gave explanations of no MIS 3 high lake level because the ¹⁴C dating method covers up as far as 50 ka. For the northern expansion of EASM, Lehmkuhl et al. (2018) also put forward that EASM could reach southern and even western Mongolia based on reconstructed lake level fluctuations. Hence, the topics in this paper are not innovative. The only attractive theme is that the high dust influx during MIS 4 is a result of the drought of megalake systems after MIS 5. Actually,

four possible mechanisms have been used to interpret this phenomenon 1) dust sources were more productive for other reasons (Rea, 1994), 2) the uplift and transport of dust were more efficient (McGee et al., 2010), 3) additional sources of dust contributed to tropical and NH fluxes and 4) depositional processes were stronger. Hence, more solid and direct evidence as the previous review should be implemented to support dust is mainly derived from the dry megalake system during MIS 4. Unfortunately, there is no such evidence in the manuscript to persuade me. On the other hand, authors ignored the fact that compared with areas of the whole Gobi desert, the area of megalake systems is still a very small portion of dust sources.

Response:

As the full titles of the publications cited in your comments are absent, we trust that the full details of these publications should be as follows in order to avoid any misunderstandings.

- Long, H., Lai, Z., Fuchs, M., Zhang, J., & Li, Y. (2012). Timing of Late Quaternary palaeolake evolution in Tengger Desert of northern China and its possible forcing mechanisms. *Global and Planetary Change*, 92, 119-129.
- Long, H., & Shen, J. (2015). Underestimated ¹⁴C-based chronology of late Pleistocene high lake-level events over the Tibetan Plateau and adjacent areas: Evidence from the Qaidam Basin and Tengger Desert. *Science China Earth Sciences*, 58(2), 183-194.
- Lai, Z., Mischke, S., & Madsen, D. (2014). Paleoenvironmental implications of new OSL dates on the formation of the “Shell Bar” in the Qaidam Basin, northeastern Qinghai-Tibetan Plateau. *Journal of Paleolimnology*, 51(2), 197-210.
- Li, G., She, L., Jin, M., Yang, H., Madsen, D., Chun, X., ... & Chen, F. (2018). The spatial extent of the East Asian summer monsoon in arid NW China during the Holocene and Last Interglaciation. *Global and Planetary Change*, 169, 48-65.
- Zhang, J., Lai, Z., & Jia, Y. (2012). Luminescence chronology for late Quaternary lake levels of enclosed Huangqihai lake in East Asian monsoon marginal area in northern China. *Quaternary Geochronology*, 10, 123-128.
- Lehmkuhl, F., Grunert, J., Hülle, D., Batkhisig, O., & Stauch, G. (2018). Paleolakes in the Gobi region of southern Mongolia. *Quaternary Science Reviews*, 179, 1-23.

- Rea, D. K., Hovan, S. A., & Janecek, T. R. (1994). Late Quaternary flux of eolian dust to the pelagic ocean. *Geomaterial Fluxes, Glacial to Recent*. National Academy Press, Washington DC.
- McGee, S. L., & Balogh, M. L. (2010). Dust accretion and destruction in galaxy groups and clusters. *Monthly Notices of the Royal Astronomical Society*, 405(3), 2069-2077.

We agree that there are many publications on the lake level change during the Quaternary across northern China and southern Mongolia, revealing a complex hydrological history in this vast area. Some of the papers mentioned in your comments have also been cited in the manuscript (e.g. Lehmkuhl et al., 2018; Lai et al., 2014; Zhang et al., 2012). However, it does not mean that our paper is less innovative or does not present new field and laboratory data supported interpretations for the following reasons:

1) The paleohydrology of Gobi Desert is still poorly known. As acknowledged by Reviewer 1 during 1st round of review, “the timing of mega lakes in the deserts of Central Asia is also a matter of debate in the recent literature”. Actually, for example, conclusions of some studies mentioned in the comment (i.e. Long et al. 2012 and Long and Shen, 2015) have already been challenged by a more recent work (Fan et al., 2020). According to the luminescence and ESR dating results, Fan et al. (2020) suggested a high lake level (~1310-1320 m) of Bajian Lake in northwestern China developed during the early stage of the Middle Pleistocene or the late stage of the Early Pleistocene, rather than MIS 5 as was reported by Long et al. (2012) and Long and Shen (2015). Thus, the statement of high lake levels during the “late Pleistocene” is still controversial. In this regard, our work on the East Gobi Desert, which is supported by multiple dating methods, provides a robust chronological control for the megalakes and therefore is of great importance for understanding the hydrological history in the Gobi Desert.

2) Studies of the paleohydrology of the East Gobi Desert are sparse and inconclusive. Most studies on high lake level chronologies (since the MIS 5) of northern China and southern Mongolia are from the western Gobi Desert (e.g. Long et al., 2012; Li et al., 2015a, 2015b, 2018; Lehmkuhl et al., 2018; Yu et al., 2019) and the Tibet Plateau (e.g. Lai et al., 2014). Their study sites are 1000-2000 km far away from our study area. We believe that our data and

analysis by our research group and others over this vast area, supports a hypothesis that climatic conditions are (and were during earlier interglacials) significantly different and the causes for high lake levels across the region are complex and incompletely understood. For example, it is suggested higher lake levels during MIS 3 in southern Mongolia were probably not related to higher precipitation values but to the higher input of melt water from the nearby Khangai Mountains (Lehmkuhl et al., 2018), whereas there is no evidence for high lake levels during MIS 3 in our study area. Additionally, lake level records since MIS 5 from the East Gobi Desert are rare and controversial. Among these studies cited in the comment, only one is from the East Gobi Desert (Zhang et al., 2012). As already stated in the manuscript (*Line 58-61 of the last round revised manuscript; Line 58-60 of this round revised manuscript*), its conclusions on MIS 3 climate conditions are different from the Wulagai Lake core records (Yu et al., 2014).

3) We reconstructed one of the largest lake systems of the Gobi Desert and derived quantitative estimates of climatic conditions of the past two interglacials from lake level and fossil records; those features distinguish our study from the works cited in the comment. The lake system reconstructed in the study, with an area of 15,500 km², is much larger than the lakes mentioned in the comment. In our study, we reconstructed the precipitation of MIS 5 and the mid-Holocene and discussed winter temperature during MIS 5. The studies cited in the comments, as far as we can determine, did not include quantitative climate reconstructions.

We note that the reviewer's comment on the four possible mechanisms for a dustier MIS 4 was proposed by Jacobel et al (2017). This is an important point and as such, we have added new paragraph to discuss this issue:

“Four hypotheses have been raised to explain the dustier MIS 4 over the northern Pacific: 1) dust sources were more productive, 2) uplift and transport of dust was more efficient, 3) additional sources of dust occurred, 4) depositional processes were stronger⁵⁴. As discussed above, the higher dust flux of MIS 4 is commonplace across East Asia and northern Pacific.

Therefore, it requires more dust availability during MIS 4 and stronger depositional processes (hypothesis 4) alone cannot explain this phenomenon. Coarser loess deposited during MIS 2 suggests stronger winds and more efficient transport of dust than during MIS 4 (Fig. 7f). Thus, hypothesis 2 is unlikely to be the explanation for a dustier MIS 4. Considering the much wetter climate across the Gobi Desert during MIS 5, drying up of the fluvial-lake system during MIS 4 might have both increased the dust productivity and added more dust sources than MIS 2. Again, as we don't have precise provenance data, it is difficult to assess the relative contribution of hypothesis 1 and 3." (*Line 531-543 of this round revised manuscript*).

As there is no detailed provenance data, we cannot determine which part of the Gobi Desert contributed more dust during MIS 4. Actually, we noted this point during the first round of revision, we already attributed a dustier MIS 4 to the humid climate across the entire Gobi Desert (not just confined to the East Gobi Desert). Additionally, lake-floor sediments are not the only dust sources; alluvial fans and fluvial channels in the desert area also contribute significantly to the dust load (Goudie and Middleton, 2006). In addition to the large lake area, the humid climate during MIS 5 also triggered formation of alluvial fans (e.g. Fig. 2f) and fluvial channels, both of which might have provided significant amount of fine particles to the dust. We note the following text (already included) on this issue:

"Humid climate across the Gobi Desert during MIS 5 likely resulted in a dustier MIS 4 over East Asia and the North Pacific." (*Line 30-31 of the last round revised manuscript; Line 30-32 of this round revised manuscript*)

"We attribute the dustier MIS 4 to a higher dust supply during MIS 4 resulting from a much more humid climate across the Gobi Desert during MIS 5, when the hydrological network in the desert was active." (*Line 487-489 of the last round revised manuscript; Line 488-491 of this round revised manuscript*)

"In addition to the megalake system in the East Gobi Desert, significant lake highstands also occurred in the central and west Gobi Desert during MIS 5^{36,59}. Fluvial/pluvial and chemical weathering processes during a humid MIS 5 in the Gobi Desert produced more fine particles

than those of the less humid MIS 3. Desiccation and shrinkage of the lakes after MIS 5 left a large area of dry lakes, alluvial fans and riverbeds in the Gobi Desert, which likely produced a large dust flux under an intensified EAWM regime and westerlies during MIS 4.” (*Line 494-500 of the last round revised manuscript; Line 496-502 of this round revised manuscript*)

Reviewer 3 questions the megalake system’s contribution to the dust as it has a small area comparing to the entire Gobi Desert. (“On the other hand, authors ignored the fact that compared with areas of the whole Gobi desert, the area of megalake systems is still a very small portion of dust sources.”)

We respectfully disagree with this comment. In addition to landform area, differences in dust emission rates are also an (or even more) important factor in determining dust flux. As we previously discussed in the manuscript, field and satellite observations have shown that dust sources are highly localized spatially and dust emission among different desert landforms can be quite different (*Line 490-494 of the last round revised manuscript; Line 494-496 of this round revised manuscript*). In addition, field measurements from Mojave Desert suggest that dust emission from the dry washes is tens of times greater than from pavements (Sweeney et al., 2011). Satellite observations from Namib Desert show that despite the fact that ephemeral lake systems and alluvial systems cover a very small proportion of the study area (2% of area) they contribute just over three quarters of observed source points (77% of plumes) (von Holdt et al., 2019). Thus, one cannot and should not assess dust emission of a landform based only on its area. Additionally, the MIS 5 megalake system in the East Gobi Desert is much larger than the lakes from central and western Gobi Desert, and therefore, dust emission from the desiccation of the East Gobi Desert megalake, cannot be underestimated. The strong lakebed deflation has been reported in other desert areas (e.g. Magee et al., 1995; Fu et al., 2017). We also find unrefuted evidence for strong deflation of the lakebed deposits in this area. We have added more text on this topic:

“The MIS 5 megalake system in the East Gobi Desert is much larger than the lakes from central and western Gobi Desert^{37, 61}. Therefore, it is possible that desiccation of the East Gobi

megalake system produced more dust than other Gobi lakes during MIS 4. We did find unrefuted evidence for the strong deflation of the lakebed deposits in this area. As shown in Supplementary Fig. 12, luminescence dating results of the lake sediments from the lowest part of NB suggest there are no lake deposits preserved at least since MIS 6, probably indicating strong deflation in this area. Formation of the shoreline and lake-floor cliffs (Supplementary Fig. 12c, d) are also probably associated with strong deflation. Considering the high lake sedimentation rate in this area, which could reach 1.6 m/ka³⁴, there would have been a significant quantity of lake sediments removed by the strong winds in this area. However, a detailed provenance study is needed to quantify the contribution of the East Gobi Desert megalake system to the higher dust flux during MIS 4.” (*Line 518-530 of this round revised manuscript*)

Revised Supplementary Fig. 12. Luminescence chronology (minimum ages) of the lake deposits from the NB. (a) Topography of the study area. **(b)** AW3D DEM data showing the sections' position and surrounding topography. Noted section 19K is located at the shoreline cliff. **(c)** Lake-floor deposits and chronology of section 19K. **(d)** Section 19M of the lake-floor cliff and its chronology. **(e)** Section 19L and its chronology.

References

Fan, Y., Li, Z., Yang, G., Yi, S., Zhang, Q., Liu, W., & Mou, X. (2020). Sedimentary evidence and luminescence and ESR dating of Early Pleistocene high lake levels of Megalake Tengger, northwestern

- China. *Journal of Quaternary Science*, **35**(8), 994-1006.
- Fu, X., Cohen, T. J., & Arnold, L. J. (2017). Extending the record of lacustrine phases beyond the last interglacial for Lake Eyre in central Australia using luminescence dating. *Quaternary Science Reviews*, **162**, 88-110.
- Goudie, A. S., & Middleton, N. J. (2006). *Desert dust in the global system*. Springer Science & Business Media.
- Jacobel, A. W., McManus, J. F., Anderson, R. F., & Winckler, G. (2017). Climate-related response of dust flux to the central equatorial Pacific over the past 150 kyr. *Earth and Planetary Science Letters*, **457**, 160-172.
- Lai, Z., Mischke, S., & Madsen, D. (2014). Paleoenvironmental implications of new OSL dates on the formation of the “Shell Bar” in the Qaidam Basin, northeastern Qinghai-Tibetan Plateau. *Journal of Paleolimnology*, **51**(2), 197-210.
- Lehmkuhl, F., Grunert, J., Hülle, D., Batkhashig, O., & Stauch, G. (2018). Paleolakes in the Gobi region of southern Mongolia. *Quaternary Science Reviews*, **179**, 1-23.
- Li, G., Jin, M., Chen, X., Wen, L., Zhang, J., Madsen, D., ... & Chen, F. (2015a). Environmental changes in the Ulan Buh Desert, southern Inner Mongolia, China since the middle Pleistocene based on sedimentology, chronology and proxy indexes. *Quaternary Science Reviews*, **128**, 69-80.
- Li, G., Jin, M., Duan, Y., Madsen, D. B., Li, F., Yang, L., ... & Chen, F. (2015b). Quartz and K-feldspar luminescence dating of a Marine Isotope Stage 5 megalake in the Juyanze Basin, central Gobi Desert, China. *Palaeogeography, Palaeoclimatology, Palaeoecology*, **440**, 96-109.
- Li, G., She, L., Jin, M., Yang, H., Madsen, D., Chun, X., ... & Chen, F. (2018). The spatial extent of the East Asian summer monsoon in arid NW China during the Holocene and Last Interglaciation. *Global and Planetary Change*, **169**, 48-65.
- Li, G., Wang, Z., Zhao, W., Jin, M., Wang, X., Tao, S., ... & Madsen, D. (2020). Quantitative precipitation reconstructions from Chagan Nur revealed lag response of East Asian summer monsoon precipitation to summer insolation during the Holocene in arid northern China. *Quaternary Science Reviews*, **239**, 106365.
- Long, H., Lai, Z., Fuchs, M., Zhang, J., & Li, Y. (2012). Timing of Late Quaternary palaeolake evolution in Tengger Desert of northern China and its possible forcing mechanisms. *Global and Planetary Change*, **92**, 119-129.
- Long, H., & Shen, J. (2015). Underestimated ¹⁴C-based chronology of late Pleistocene high lake-level events over the Tibetan Plateau and adjacent areas: Evidence from the Qaidam Basin and Tengger Desert. *Science China Earth Sciences*, **58**(2), 183-194.

- Magee, J. W., Bowler, J. M., Miller, G. H., & Williams, D. L. G. (1995). Stratigraphy, sedimentology, chronology and palaeohydrology of Quaternary lacustrine deposits at Madigan Gulf, Lake Eyre, South Australia. *Palaeogeography, Palaeoclimatology, Palaeoecology*, **113**(1), 3-42.
- Sweeney, M. R., McDonald, E. V., & Etyemezian, V. (2011). Quantifying dust emissions from desert landforms, eastern Mojave Desert, USA. *Geomorphology*, **135**(1-2), 21-34.
- Von Holdt, J. R. C., Eckardt, F. D., Baddock, M. C., & Wiggs, G. F. (2019). Assessing landscape dust emission potential using combined ground-based measurements and remote sensing data. *Journal of Geophysical Research: Earth Surface*, **124**(5), 1080-1098.
- Yu, K., Lehmkuhl, F., Schlütz, F., Diekmann, B., Mischke, S., Grunert, J., ... & Zeeden, C. (2019). Late Quaternary environments in the Gobi Desert of Mongolia: Vegetation, hydrological, and palaeoclimate evolution. *Palaeogeography, Palaeoclimatology, Palaeoecology*, **514**, 77-91.
- Yu, Z., Liu, X., Wang, Y., Chi, Z., Wang, X., & Lan, H. (2014). A 48.5-ka climate record from Wulagai Lake in Inner Mongolia, Northeast China. *Quaternary International*, **333**, 13-19.
- Zhang, J., Lai, Z., & Jia, Y. (2012). Luminescence chronology for late Quaternary lake levels of enclosed Huangqihai lake in East Asian monsoon marginal area in northern China. *Quaternary Geochronology*, **10**, 123-128.

[Comment 3]

In reply to reviewer 3#, I really can't give positive feedback on lake evaporation. Firstly, the authors proposed that modern evaporation rates in the south of the East Gobi Desert areas can be surrogates for the evaporation rates of the study area during MIS 5 and mid-Holocene. However, the coldest month means temperature during MIS 5 would have been over 10 °C warmer than at present based on temperature reconstructions. Such large-amplitude changes in temperature certainly have a substantial influence on lake evaporation during MIS 5 relative to MIS 1 and current conditions. Secondly, the authors inferred evaporation rate probably also moved northward during those two interglacials across East Asia. In Fig. 12, distributions of PET over China didn't favor a northward displacement of evaporation rate when precipitation intensity moved northward. This is because there is no regular pattern of evaporation rate fluctuations in China.

Response:

We agree that temperature is an important factor influencing evaporation. However, we do not think evaporation can be assessed based only on temperature. As we already stated in the manuscript, there are many factors determining evaporation (*Line 728-729 of the last round revised manuscript; Line 755-756 of this round revised manuscript*). The evaporation process is primarily driven by two factors: (1) a source of energy to supply the latent heat of vaporization, and (2) a concentration gradient in the water vapor, typically provided by air movement (Ward et al., 2015). In addition to temperature, lake evaporation is also dependent on other variables, such as humidity, wind speed, and net radiation (e.g. Friedrich et al., 2018). Thus, as shown by the following figure (**revised supplementary Fig. 13**), the desert area has a higher PET than South China although the desert area has a much lower temperature. **A higher temperature is not necessarily associated with a higher evaporation.**

Based on our results MIS 5 has a much higher temperature than the present, which would tend to increase the evaporation. However, on the other hand, wind speed was much lower during MIS 5 and mid-Holocene as indicated by grain size record of loess deposits (e.g. Ding et al., 2002). Also, high lake levels during MIS 5 and mid-Holocene suggest a likely higher than present atmosphere humidity. Both the lower wind speed and higher humidity could cause a decline in evaporation (McVicar et al., 2012). Thus, it is difficult to estimate the evaporation based only on temperature.

Instead, we used a relatively wide range of evaporation (900-1100 mm/year) in the model to deal with the uncertainties. We believe it is a reasonable approximation to the possible range of values the region experienced. As shown in the following figure, the evaporation rate of most of South and East China falls within this range. It should be noted that some parts of South China, where the January temperature is 20-25 °C higher than the study area, also have an annual PET of 900-1100 mm. Therefore, although the coldest monthly temperature in the East Gobi Desert would have been 10 °C higher than the present during MIS 5, the evaporation rate was still unlikely out of the range (i.e. 900-1100 mm/year).

Revised Supplementary Fig. 13. Mean annual potential evapotranspiration (PET, a) and mean January temperature (b) over the East Gobi Desert and surrounding area. Note that the area with mean annual PET between 900-1100 mm is marked in gray. Data source: The temperature data (1970-2000) is from WorldClim Version 2.0, available at <http://www.worldclim.com/version2>; the PET data (1981-2010) is from TerraClimate (Abatzoglou, J. et al., 2015).

The reviewer also mentioned “the authors inferred evaporation rate probably also moved northward during those two interglacials across East Asia”. We are grateful to the reviewer for pointing out this issue. This was poorly addressed in the last response letter. What we wanted to show is that the climate zone would have moved northward during interglacials. Therefore, the modern climate of South and East China would be an analogue of the interglacial climate of the East Gobi Desert. Accordingly, evaporation in the East Gobi Desert during interglacials should be similar to the modern evaporation in East and South China, where the climate is warmer and wetter than the modern East Gobi Desert.

In order to clarify this issue, we have rewritten this section of text as follows:

“Factors affecting potential evapotranspiration (PET) include solar radiation, temperature, wind velocity, relative humidity, etc. Modern mean annual PET in the study area is 1060 mm (Supplementary Fig. 13a). Due to drier climate and higher wind speeds, desert areas in northwest China have the highest PET, which may exceed 1500 mm/yr. PET is 900-1100

mm/yr in most of South and East China (Supplementary Fig. 13a). MIS 5 has a much higher temperature than the present, which tends to increase the evaporation. On the other hand, however, the wind was much weaker during MIS 5 and mid-Holocene as indicated by the grain size records of the loess deposits (Fig. 7). Also, high lake levels during MIS 5 and mid-Holocene across the study areas suggest a probably higher than the present atmospheric humidity. Both the lower wind speed and higher humidity would have offset the increasing evaporation from the warming temperature. Thus, it is difficult to estimate the evaporation based only on temperature.

Considering the northward displacement of the climate zone during MIS 5 and the mid-Holocene, it is reasonable to use modern PET of East and South China, which is 900-1100 mm/yr, as an approximation to the lake evaporation of East Gobi Desert during MIS 5 and the mid-Holocene. We believe this relatively wide range that could deal with uncertainties caused by the different climate conditions. It is worth to note that some parts of South China where the January temperature is 20-25 °C higher than the study area also have an annual PET of 900-1100 mm (Supplementary Fig. 13). Therefore, although the coldest monthly temperature in the East Gobi Desert would have been 10 °C higher than the present during MIS 5, the evaporation rate during MIS 5 is still unlikely out of the range (i.e. 900-1100 mm/yr). In the water balance model, three different lake evaporation rates, i.e., 900 mm/yr, 1000 mm/yr and 1100 mm/yr, are used to estimate precipitation during MIS 5 and the mid-Holocene (Fig. 6a-c).” (*Line 755-778 of this round revised manuscript*)

References

- Ding, Z. L., Derbyshire, E., Yang, S. L., Yu, Z. W., Xiong, S. F., & Liu, T. S. (2002). Stacked 2.6-Ma grain size record from the Chinese loess based on five sections and correlation with the deep-sea $\delta^{18}\text{O}$ record. *Paleoceanography*, **17**(3), doi:10.1029/2001PA000725.
- Friedrich, K., Grossman, R. L., Huntington, J., Blanken, P. D., Lenters, J., Holman, K. D., ... & Kowalski, T. (2018). Reservoir evaporation in the Western United States: current science, challenges, and future needs. *Bulletin of the American Meteorological Society*, **99**(1), 167-187.
- McVicar, T. R., Roderick, M. L., Donohue, R. J., Li, L. T., Van Niel, T. G., Thomas, A., ... & Dinpashoh,

Y. (2012). Global review and synthesis of trends in observed terrestrial near-surface wind speeds: Implications for evaporation. *Journal of Hydrology*, **416**, 182-205.

Ward, A. D., Trimble S. W., Burckhard, S. R. & Lyon, J. G. (2015). *Environmental Hydrology*. CRC Press.

[Comment 4]

Line 28, explain the corresponding period, whole MIS 5, or specific period.

Response:

According to the numerical dating results, the megalake system probably occurred during MIS 5e or lasted even longer. However, considering that the minimum uncertainty on a luminescence age is 5% (at 1 σ ; i.e at least 6 ka at 125 ka), it is difficult to confine the specific period of the high lake level with confidence. We are limited by the precision of our dataset.

[Comment 5]

Line 34-35, the Authors emphasize that their study is different from a speleothem from caves in southern China. Why is specially referred to as southern China instead of the whole of China? There also have been reported speleothem caves records in northern and southwestern China during MIS 5.

Response:

Thank you for your comment. We agree that there are also many stalagmite records in other parts of China, especially southwestern China. However, it is shown that $\delta^{18}\text{O}$ records from the different areas (over 1000 km apart from each other) always replicate precisely during contemporary periods (e.g. Yuan et al., 2004; Cheng et al., 2019). Additionally, cave records from northern China are younger than MIS 5 (e.g. Tan et al., 2003; Duan et al., 2016; Jia et al., 2022) or less continuous (e.g. Li et al., 2020), making them only peripherally useful for MIS 5

analysis. As the stalagmite records were not discussed much in the manuscript, we have deleted those words from the abstract during revision.

Reference

Cheng, H., Zhang, H., Zhao, J., Li, H., Ning, Y., & Kathayat, G. (2019). Chinese stalagmite paleoclimate researches: A review and perspective. *Science China Earth Sciences*, **62**(10), 1489-1513.

Duan, W., Cheng, H., Tan, M., & Edwards, R. L. (2016). Onset and duration of transitions into Greenland Interstadials 15.2 and 14 in northern China constrained by an annually laminated stalagmite. *Scientific Reports*, **6**(1), 1-6.

Jia, W., Zhang, P., Wang, X., Cheng, H., He, S., Shi, H., ... & Edwards, R. L. (2022). Chinese Interstadials 14–17 recorded in a precisely U-Th dated stalagmite from the northern edge of the Asian summer monsoon during the MIS 4/3 boundary. *Palaeogeography, Palaeoclimatology, Palaeoecology*, **607**, 111265.

Li, Y., Rao, Z., Xu, Q., Zhang, S., Liu, X., Wang, Z., ... & Chen, F. (2020). Inter-relationship and environmental significance of stalagmite $\delta^{13}\text{C}$ and $\delta^{18}\text{O}$ records from Zhenzhu Cave, north China, over the last 130 ka. *Earth and Planetary Science Letters*, **536**, 116149.

Tan, M., Liu, T., Hou, J., Qin, X., Zhang, H., & Li, T. (2003). Cyclic rapid warming on centennial-scale revealed by a 2650-year stalagmite record of warm season temperature. *Geophysical Research Letters*, **30**, 1617.

Yuan, D., Cheng, H., Edwards, R. L., Dykoski, C. A., Kelly, M. J., Zhang, M., ... & Cai, Y. (2004). Timing, duration, and transitions of the last interglacial Asian monsoon. *Science*, **304**(5670), 575-578.

[Comment 6]

Line42-44, how much area of the Gobi Desert?

Response:

While there is no consensus on the area of the Gobi Desert. Sternberg et al. (2015) and other scientific literature lists the Gobi Desert as 1,160,000-1,700,000 km² in extent.

Reference

Sternberg, T., Rueff, H., & Middleton, N. (2015). Contraction of the Gobi Desert, 2000–2012. *Remote Sensing*, 7(2), 1346-1358.

[Comment 7]

Line 64-65, the Authors mean geomorphological process rather than monsoon precipitation influenced hydrological conditions.

Response:

We respectfully disagree with your understanding on the paper by Yang et al. (2015). Although the authors emphasized the importance of the geomorphological processes in the desertification and hydrological conditions changes of Hunshandake Sandy Land, the impact of climate change on desertification was not ruled out. The following is the original text of the paper (Yang et al., 2015)

“Well-developed dark grassland-type paleosols (mollisols) at the southern edge of the Hunshandake, OSL-dated to between 6.93 ± 0.61 and 4.27 ± 0.38 ka (Fig. 3), also suggest a wetter climate. Lacustrine sands underlying this paleosol indicate an earlier wetland environment followed by soil formation that indicates a rapid transition to dry conditions at ca. 4.2 ka. **Because the southern part of the Hunshandake was not impacted by ground water sapping until recently**, it returned to green conditions again at ca. 2.8 ka and maintained this state for slightly longer than 1,000 y (Fig. 3, section P).” (*Page 704*)

Well-developed dark grassland-type paleosols (mollisols) at the southern edge of the Hunshandake, OSL-dated to between 6.93 ± 0.61 and 4.27 ± 0.38 ka (Fig. 3), also suggest a wetter climate. Lacustrine sands underlying this paleosol indicate an earlier wetland environment followed by soil formation that indicates a rapid transition to dry conditions at ca. 4.2 ka. Because the southern part of the Hunshandake was not impacted by ground water sapping until recently, it returned to green conditions again at ca. 2.8 ka and maintained this state for slightly longer than 1,000 y (Fig. 3, section P). Similarly, paleosols developed during the period between ca. 9.6 and ca. 3 ka in the western part of Hunshandake (6), where groundwater sapping has yet to occur. The total area of desertification since ca. 4.2 ka ago is $>20,000$ km² based on field observations and associated mapping.

“The Hunshandake green/desert switch, similar to that found in the Sahara, **occurred in conjunction with regional drying**, as monsoonal flow decreased over the region (6, 8).” (Page 705)

ditions between 8 and 4 ka, as well as noticeable decline in tree pollen at ca. 4 ka, associated with large-scale drying (8, 13).

The Hunshandake green/desert switch, similar to that found in the Sahara, occurred in conjunction with regional drying, as monsoonal flow decreased over the region (6, 8). However, unlike the Sahara, where a return to a green state may occur under a scenario with future wetter conditions and increased surface vegetation (31), reestablishment of a green Hunshandake is

“The combination of factors at ca. 4.2 ka, including fluvial degradation by ~30 m and a similar drastic lowering of the water table, which led to the disappearance of lakes, **coupled with a precipitous drop in monsoonal precipitation**, resulted in a dominance of aeolian processes.” (Page 705)

hysteresis. The combination of factors at ca. 4.2 ka, including fluvial degradation by ~30 m and a similar drastic lowering of the water table, which led to the disappearance of lakes, coupled with a precipitous drop in monsoonal precipitation, resulted in a dominance of aeolian processes. This produced the extreme aridification of the Hunshandake observed since 4.2 ka. We note that

Reference

Yang, X., Scuderi, L. A., Wang, X., Scuderi, L. J., Zhang, D., Li, H., ... & Yang, S. (2015). Groundwater sapping as the cause of irreversible desertification of Hunshandake Sandy Lands, Inner Mongolia, northern China. *Proceedings of the National Academy of Sciences*, **112**(3), 702-706.

[Comment 8]

Line 82-83, in Fig. 1a, three red arrows array seemingly indicate the moisture is from Bengal bay. If so, it is ISM, not EASM.

Response:

We disagree with you on this point. The red arrows in Fig. 1a are schematic representation of near surface and lower troposphere winds, which carry water vapor during summer in this area. As shown by the following figure and reported extensively in the literature, a large quantity of water vapor is transported from the Indian Ocean to South China Sea and China during summer (e.g. Wang et al., 2003; Ding and Chan, 2005; Zhou and Yu; 2005; An et al., 2015). The circulation of the East Asian summer monsoon (EASM) is characterized by strong cross-equatorial flows in the lower troposphere as well as a strong westerly flow over South India and southwesterly flow over China (Lau and Li, 1984). Thus, the water vapor and wind indicated by the red arrow in Fig. 1a is a component of the EASM. The extent of different monsoons systems is always defined by the geographic locations rather than the moisture source. Wang et al. (2003) defined the Indian monsoon over sector 40°-105°E and the EASM over the 105°-160°E. Wang and LinHo (2002) suggested the Indian Summer Monsoon was located west of 100°E. Although there is disagreement on whether to include the South China Sea in the EASM (Ding, 1994) or the western North Pacific summer monsoon (WNPSM. Wang and LinHo, 2002), the South China Sea and South China are not treated as a part of the ISM.

Vertically integrated climate mean (1951–1999 average) JJA water vapor transport ($\text{kg} \times \text{m}^{-1} \text{s}^{-1}$).

The coloring indicates the magnitude of the moisture flux vector (after Zhou and Yu, 2005)

Reference

An, Z., Wu, G., Li, J., Sun, Y., Liu, Y., Zhou, W., ... & Feng, J. (2015). Global monsoon dynamics and climate change. *Annual Review of Earth and Planetary Sciences*, **43**, 29-77.

Ding, Y. (1994). *Monsoons over China*. Springer Science & Business Media Dordrecht, New York.

Ding, Y., & Chan, J. C. (2005). The East Asian summer monsoon: an overview. *Meteorology and Atmospheric Physics*, **89**(1), 117-142.

Lau, K. M., & Li, M. T. (1984). The monsoon of East Asia and its global associations—A survey. *Bulletin of the American Meteorological Society*, **65**(2), 114-125.

Wang, B., Clemens, S. C., & Liu, P. (2003). Contrasting the Indian and East Asian monsoons: implications on geologic timescales. *Marine Geology*, **201**(1-3), 5-21.

Wang, B. & LinHo (2002). Rainy season of the Asian–Pacific summer monsoon. *Journal of Climate*, **15**(4), 386-398.

Zhou, T. J., & Yu, R. C. (2005). Atmospheric water vapor transport associated with typical anomalous summer rainfall patterns in China. *Journal of Geophysical Research: Atmospheres*, **110**, D08104.

[Comment 9]

Line 466-468, As authors firstly cited Kug et al. (2015), a warming Arctic is responding to harsh winters in the East Asia area. However, the authors then proposed that Arctic warmth might have coexisted with warming water based on fluminea occurrence and abundance at mid-latitude East Asia during MIS 5. This is illogical.

Response:

We do not see that there is any illogical discussion/conclusion on this issue. Kug et al. (2015) believed severe winters across northern mid-latitudes were associated with anomalous warmth of different part of Arctic Ocean based on modern observational analyses and model results. **That is Kug et al.'s (2015) view, not ours.** What we expressed in the manuscript is that the East Gobi Desert did not record this phenomenon, i.e. warm Arctic while harsh winters over mid-latitudes, during MIS 5 high lake level period. Our study suggests Arctic warmth might have coexisted with a warming mid-latitude East Asia during MIS 5; Arctic warmth is not necessarily accompanied with a harsh winter over northern mid-latitudes. We do not think that a discussion on different views from different studies in a paper is illogical.

Reference

Kug, J. S., Jeong, J. H., Jang, Y. S., Kim, B. M., Folland, C. K., Min, S. K., & Son, S. W. (2015). Two distinct influences of Arctic warming on cold winters over North America and East Asia. *Nature Geoscience*, **8**, 759-762.

[Comment 10]

Line 411-412, please add a reference

Response:

Done.

[Comment 11]

The data for regression between runoff coefficient and mean annual precipitation for 92 modern catchments are from north of China where is outside the study area (Fig. 6d). It is reasonable for me to infer that regression results can't reflect the true conditions for the study area.

Response:

As the reviewer already mentioned in his/her "Comment 3", the climate during MIS 5 was quite different from the present day climate. Accordingly, hydrological conditions over the East Gobi Desert during MIS 5 were also different when compared to the present (Fig. 1b and Fig. 4b). In fact, there is little in the way of perennial streams over the most of the study area at present (Fig. 1b). Therefore, we cannot use modern precipitation-runoff coefficient regression results of the study area to estimate the precipitation of MIS 5.

We believe this regression result is reasonable for the following reasons:

Firstly, the modern climate conditions of the 92 catchments are similar to that of the study area during interglacials. The 92 catchments are located south of the study area; they have a higher precipitation than the study area (Fig. 6d). We believe, considering the northward displacement of climate zones during MIS 5 and the mid-Holocene, that the modern climate and hydrological conditions of the 92 catchments are analogous to that of the study area during MIS 5 and the mid-Holocene.

Secondly, the strong correlation between the runoff coefficient and mean annual precipitation ($p < 0.0001$, Supplementary Fig. 10) suggests the regression equation is robust and applicable to a large area rather than a specific catchment. As shown in Fig. 6d, the 92 catchments are spread over an area ~1200 km in the north-south direction and ~800 km in the east-west direction. Therefore, we believe the regression obtained from this vast area also works well in the study area during interglacials.

Thirdly, and more importantly, as shown in Supplementary Fig. 10, our regression equation works well in the study area. The mid-Holocene runoff coefficient-precipitation

data from two independent studies from Chagan Nur Lake and Dali Lake (Li et al., 2020; Goldsmith, et al., 2017), both of which are located in the East Gobi Desert, fall within the 95% confidence interval of the regression equation (Supplementary Fig. 10). These results again confirm the robustness of the regression equation and its applicability for the study area.

Supplementary Fig. 10. Regression between runoff coefficient (c) and mean annual precipitation (P) for 92 modern catchments of North China, with 95% confidence interval shown by the green lines. The precipitation and runoff coefficients of Chagan Nur¹³ and Dali Lake¹⁴ during Holocene high-stand periods, both of which are located in the East Gobi Desert, are also plotted (blue circle and orange square, respectively). The regression line with 95% confidence level contains the two lakes data, indicating the empirical equation works well in the East Gobi Desert. Data source: Yang et al. (2007)¹⁵.

Reference

Goldsmith, Y., Broecker, W. S., Xu, H., Polissar, P. J., Demenocal, P. B., Porat, N., ... and An, Z. (2017). Northward extent of East Asian monsoon covaries with intensity on orbital and millennial timescales.

Proceedings of the National Academy of Sciences, **114**(8): 1817-1821.

Li, G., Wang, Z., Zhao, W., Jin, M., Wang, X., Tao, S., ... and Madsen, D. (2020). Quantitative precipitation reconstructions from Chagan Nur revealed lag response of East Asian summer monsoon precipitation to summer insolation during the Holocene in arid northern China. *Quaternary Science Reviews*, **239**: 106365.

Yang, D., Sun, F., Liu, Z., Cong, Z., Ni, G., and Lei, Z. (2007). Analyzing spatial and temporal variability of annual water-energy balance in nonhumid regions of China using the Budyko hypothesis. *Water Resources Research*, **43**(4): W04426.